# The functional role of visual information and fixation stillness in the quiet eye

**David J. Harris** [ID]*, Mark R. Wilson, Samuel J. Vine

School of Sport and Health Sciences, University of Exeter, Exeter, United Kingdom

* d.j.harris@exeter.ac.uk

## Abstract

The final fixation to a target in far-aiming tasks, known as the *quiet eye*, has been consistently identified as an important perceptual-cognitive variable for task execution. Yet, despite a number of proposed mechanisms it remains unclear whether the fixation itself is driving performance effects or is simply an emergent property of underpinning cognitions. Across two pre-registered studies, novice golfers ($n$ = 127) completed a series of golf putts in a virtual reality simulation to examine the function of the quiet eye in the absence of visual information. In experiment 1 participants maintained a quiet eye fixation even when all visual information was occluded. Visual occlusion did significantly disrupt motor skill accuracy, but the effect was relatively small (89cm vs 105cm radial error, std. beta = 0.25). In experiment 2, a 'noisy eye' was induced using covertly moving fixation points, which disrupted skill execution ($p$ = .04, BF = 318.07, std. beta = -0.25) even though visual input was equivalent across conditions. Overall, the results showed that performers persist with a long pre-shot fixation even in the absence of visual information, and that the stillness of this fixation confers a functional benefit that is not merely related to improved information extraction.

## General introduction

In visually-guided motor tasks such as catching a ball, threading a needle, or aiming a projectile at a target, the way in which visual attention is deployed, spatially and temporally, is an important determinant of the accuracy of the subsequent motor response [1–3]. Much research has addressed one particular instance of this effect, known as the 'quiet eye', wherein the duration of the final fixation to a target location, prior to movement execution, is an important determinant of subsequent performance in far-aiming tasks [2, 4–6]. The duration of the quiet eye fixation is proposed to be a determinant of performance success [6]; it is longer in successful compared to unsuccessful attempts and in experts compared to [4]. Yet, despite many studies on the quiet eye phenomenon, the exact functional mechanism–that is, the way in which quiet eye supports skilled performance–remains unclear [5, 7, 8].

### Mechanisms of the quiet eye

Several competing accounts have been proposed to explain the function of the quiet eye but none have successfully accounted for all existing findings. Vickers [9] suggests that the quiet

**Data Availability Statement:** All analysis scripts and raw data are available from the Open Science Framework: https://osf.io/35fgp.

**Funding:** DH's time was supported by a Royal Academy of Engineering UKIC Fellowship. The

funders had no role in study design, data collection and analysis, decision to publish, or preparation of the manuscript.

**Competing interests:** The authors have declared that no competing interests exist.

eye fixation facilitates information processing and that the fixation duration reflects the time needed to programme a motor response (known as the *response programming* explanation; see also [10, 11]). Alternatively, Klostermann and colleagues have proposed an *inhibition hypothesis* where the quiet eye serves to inhibit sub-optimal motor responses in favour of the optimal variant [12, 13]. Further, quiet eye has also been linked to *attentional control* and *attentional focus* functions [14, 15], where it is proposed to facilitate the processing of goal-relevant, and inhibition of goal-irrelevant, stimuli [2, 15]. Yet, despite this range of proposed explanations no single account can comprehensively explain the range of quiet eye findings, such as sensitivity of the fixation to anxiety [16], changing task demands [17], and target location pre-cueing [18]. Indeed, recent work has also questioned the putative close coupling between quiet eye and performance, finding that experimental manipulations of the duration and location of the quiet eye fixation had little effect on performance [19]. Although, Sun et al. [20] and Klostermann et al. [21] found greater impacts of manipulating task constraints that shortened the quiet eye duration in tasks requiring interception of moving targets. Consequently, it remains unclear what might be driving long pre-movement fixations.

Allocation of gaze in time and space is generally proposed to provide a real-time index of the information-processing priorities of the visual system–commonly referred to as the "eye-mind link" [22]. However, eye movements can also reflect internal processes, and properties related to the eyes (e.g., saccade rate, pupil size) have been widely used to index global brain states (e.g., arousal, motivation and cognitive effort) [23, 24]. Further to this, during perceptually decoupled states of attention–states such as mind wandering–external information processing can become deprioritised and eye movement behaviours no longer serve an information processing function but are instead driven by internal states [25, 26]. So the quiet eye fixation is not necessarily a direct indicator of information processing priorities, as is often assumed.

Even if eye movements are not always performing an information acquisition role, they may still provide functional benefits. For instance, fixations to locations where objects have previously appeared are thought to aid in memory retrieval [27, 28] as well as mental concentration [29]. Alternatively, the cessation of actively seeking visual information is potentially adaptive because it could prevent unimportant external events from disrupting internal processes [30]. In summary, gaze behaviours do not always reflect a search for information, but instead may be an epiphenomena of ongoing cognition, or even facilitate spatial memory or active suppression of external information. The implication for the quiet eye is that the fixation could be driven by internal states, rather than the search for information, yet still maintain some functional role.

## Outward-in and inward-out functions

In order to better describe and understand the function of the quiet eye, we make a distinction between two broad roles that the pre-movement fixation could play; roles which have implications for the nature of the causal relationship between the fixation and the performance outcome. For simplicity we will refer to these roles as 'outward-in' and 'inward-out' (as described in [19]). On the one hand, the quiet eye fixation might perform an *outward-in* role where the duration and location of the fixation, and the visual information gathered therein, are a major determinant of ongoing cognition and action planning/execution–reflecting the "eye-mind link" and a more active sampling of the visual scene. From this perspective–which aligns with traditional response programming explanations–disruption of the fixation would also disrupt performance, because the fixation (where it is located and for how long) are directly informing action planning/execution.

Alternatively, an *inward-out* perspective on quiet eye suggests that, in many instances, the fixation is merely *reflective* of ongoing cognitive processes (e.g., focusing of attention, motor planning) that may in turn influence performance, rather than performing any useful information processing function itself. The inward-out perspective could be further subdivided into two levels: either the fixation is an epiphenomenon of ongoing cognition and serves no function, or the fixation provides some functional benefit that is *not* information acquisition. Falling into the former are instances in which a focused performer tends to display longer pre-shot fixations because their attention is goal-directed, but they do not actually *require* a long fixation (as seems to be the case for experts). In this instance, the individual is likely to perform well, not because of the specific parameters of the quiet eye fixation, but because they were generally focused and attentive [31–33]. Falling into the latter, is a functional suppression of visual information through maintenance of a single fixation [30]. In support of this hypothesis is the finding that visual processing actually appears to be suppressed during the quiet eye [34]. To date, however, no experimental work has tried to disambiguate these outward-in and inward-out functions of the quiet eye.

It is important to note that the functional role played by the quiet eye is likely to vary considerably between tasks, particularly in relation to the amount of preparation time that is available [8]. In self-paced tasks (e.g., throwing a dart at a dartboard) a series of fixations can be made to support the planning of the upcoming movements and hence the final fixation is likely to be less important for extracting visual information. However, in externally-paced tasks (e.g., catching a ball) the skill may depend entirely on a single fixation or tracking gaze for identifying the location and motion of the target [35, 36]. In the latter case, it is inevitable that the quiet eye takes on a more critical outward-in function. Yet it is unclear whether the same is true for skills in which the final fixation is not critical for acquiring the location of the target. For instance, Horn and Marchetto [18] have recently shown that a higher degree of certainty about the upcoming target location reduces the pre-programming role of the quiet eye (but see also [37]). Consequently, in the present work we adopted the skill of golf putting, perhaps the most studied self-paced skill in quiet eye research (see Lebeau et al. [4] meta analysis and review), to explore whether the quiet eye maintains an outward-in role when ample preparation time is available.

## The present study

In this work we present two novel quiet eye manipulations, made possible by studying perception and action in VR, that can be used to explore whether quiet eye plays more of an inward-out or outward-in function during a self-paced skill (the golf putt). In experiment 1 we explored whether the quiet eye fixation performs a function beyond just visual processing by testing whether performers persist in maintaining a long and stable pre-shot fixation when visual information is occluded. In experiment 2, we introduce a novel manipulation that disrupted the stillness of the quiet eye to explore the significance of maintaining a still fixation, irrespective of visual input. Answers to these mechanistic questions about the quiet eye are fundamental to a better understanding of the perception-action process in far aiming tasks, as well as for designing effective interventions.

## Experiment 1 –Visual occlusion

Experiment 1 aimed to test outward-in explanations of the quiet eye. If the principal function of the pre-shot fixation is to collect visual information and orient the performer for the upcoming shot, then a long fixation would not be required, or be useful, in the absence of visual information. As we had no a priori hypothesis about which elements of the scene were

important, everything was occluded. Hence, we aimed to test 1) if performers maintained a long stable fixation when visual information was occluded, and 2) if longer fixations were still related to better performance when vision was occluded. A putting condition was created in which the participant was able to briefly view the hole and ball (for 1.5 seconds) to orient themselves to the putt, after which all visual information was occluded. If the quiet eye primarily serves to gather environmental information and guide putter-ball contact (outward-in explanation), there would be no use for a long fixation in this instance and quiet eye durations should be shorter in the occlusion condition. By contrast, an inward-out interpretation would predict that stable fixations might still be made if the quiet eye is not just for acquiring visual information but is linked to other ongoing cognitive processes, such as a functional decoupling of perception or supporting neural quiescence. In line with this interpretation, it was also predicted that longer quiet eye durations would still be related to improved performance in the occluded vision condition.

## Experiment 1 –Method

### Preregistration

The research question, hypotheses, sampling plan, methods, materials, and statistical analyses were all pre-registered on the Open Science Framework and can be accessed online (https://osf.io/vrbcf/). Any additional analyses not present in the preregistration are specified in the analyses as exploratory.

### Design

A repeated measures design was used, with participants completing a free putting condition and the occluded condition in a counterbalanced order. The primary outcome measure was quiet eye duration (in milliseconds) and secondary measures were putting accuracy (radial error in cm) and putting kinematics (accelerations of the clubhead in x,y,z).

### Participants

47 participants (15 females), all novice golfers, were recruited using convenience sampling from the University undergraduate population. Qualification as a novice was based on having no official golf handicap or prior formal golf putting experience (as in [38]). We chose to sample from a novice population to enable this larger sample size, and while novices do tend to exhibit greater variation in both quiet eye and performance, the golf putt was deemed to be a sufficiently simple aiming skill that novices could still perform it adequately. Sample size was determined using the "SIMR" package for R [39]. The study was powered to find the smallest meaningful effect of interest, which was set at a 75ms change in quiet eye duration. The 75ms value, while slightly arbitrary, was selected because a change this size would be considered very small in relation to previous literature and typical variances in QE values among novices [40, 41]. Monte Carlo simulations ($n = 1000$) of a series of linear mixed effects models were run under scenarios of increasing sample size using SIMR to generate a power curve. Participant was set as a random factor, and $\beta = 75.0$. Given 20 trials per participant, 85% power was reached for a sample size of $\sim 45$ (the power curve for the analysis is available in the supplementary materials: https://osf.io/56fmv/). Participants were provided with details of the study and gave written informed consent on the day of the testing visit. Ethical approval (REF: 191023-A-08) was obtained from the School of Sport Science Research Ethics Committee prior to data collection. The study, and collection of data, took place between November 2019

and March 2020. Authors had access to participants' identifying information during the data collection, but this was destroyed when the study was completed.

## Task and materials

**VR golf putting.**    The VR golf putting simulation was developed using the gaming engine Unity 2019.2.12 (Unity technologies, CA) and C#. Graphics were generated on an HP Elite-Desk PC running Windows 10, with an Intel i7 processor and Titan V graphics card (NVIDIA Corp., Santa Clara, CA). The construct validity of an earlier version of this task for simulated putting has previously been tested and supported (see [19, 42] for more details of the simulation validation). The putting simulation was displayed using the HTC-Vive (HTC, Taiwan), a 6-degrees-of-freedom, consumer-grade VR-system with 110˚ field of view that has been validated for movement task research [43] (see Fig 1). The Vive headset featured built in Tobii eye tracking, which uses binocular dark pupil tracking and samples at 120Hz across the full 110˚ field of view of the HMD, to an accuracy of 0.5˚. Gaze was calibrated in VR over 5 points prior to each block of putts and the experimenter further checked the accuracy by asking the participant to fixate on the ball. The accuracy of both the Vive eye tracking system [44] and the head and controller position tracking [43] have been previously validated. The VR putter was created and tracked by attaching a Vive sensor to the head of a real golf club (position sampled at 90Hz, accuracy 1.5cm). Participants putted from 10ft (3.05m) to a target the same size and shape (diameter 10.80cm) as a standard hole (as in [45]). Participants were instructed to land the ball as close as possible to the target, but the ball did not drop into the hole. Auditory feedback mimicking the sound of a club striking a ball was provided concurrent to the visual contact of the club head with the ball. The game also included ambient environmental noise to create realism and immersion.

In the visual occlusion condition participants were able to preview the hole for 1.5 seconds which allowed a brief opportunity to align themselves with the ball and target. After 1.5 seconds all relevant visual information was occluded by rendering a virtual white sphere around the head of the participant so that all they could see was the colour white. The 1.5s duration was chosen based on pilot testing, which showed this was sufficient time for participants to

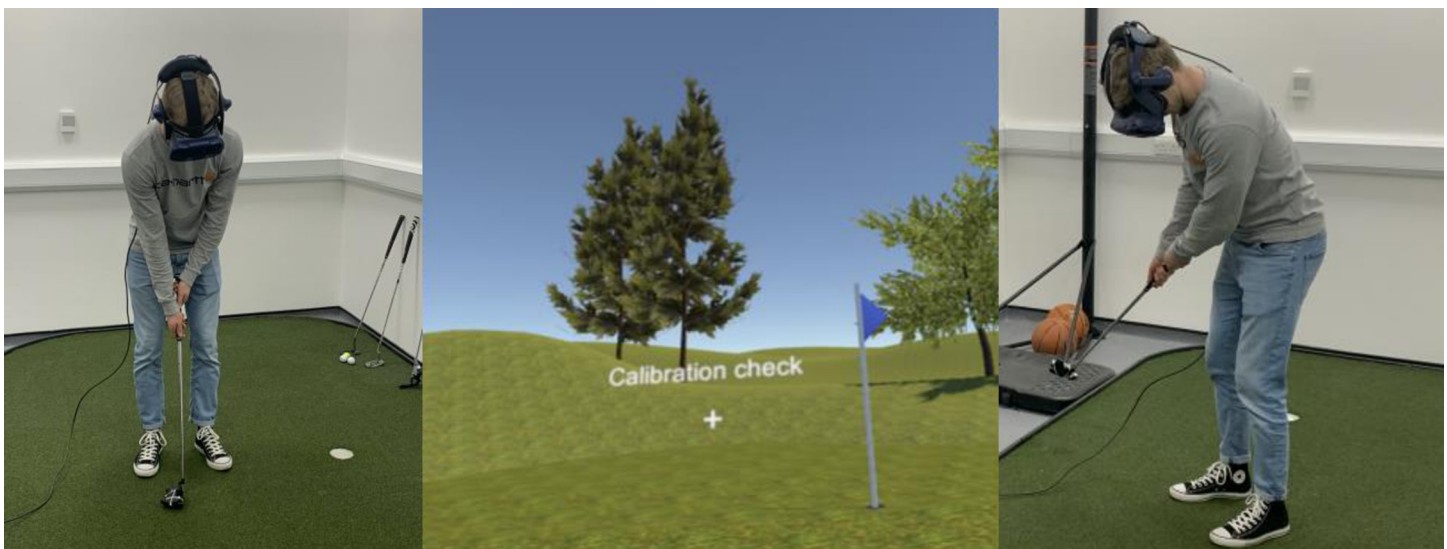

**Fig 1. Golf putting task.** The golf putting task illustrating the headset and tracked putter (left and right images) and screenshot from the VR environment (centre).

orient themselves to the task, but not enough time to actually perform the putt. Consequently, the putt was always performed with the occlusion present. The sphere remained during the putting movement and was removed 1 second after contact with the ball had been made, so that the participant could view the final location of their putt.

## Measures

**Quiet eye period.** The quiet eye period was operationally defined as the final fixation directed toward the ball which was initiated prior to the critical movement, i.e. the backswing of the club [6, 38]. Quiet eye fixations had to begin prior to the critical movement, but could continue throughout the movement, provided they remained on the ball (e.g. as in [46]). An automated method of quiet eye analysis in MATLAB R2018a (Mathsworks, MA) was used (as in [19, 47]). Gaze data were initially denoised using a second-order lowpass (30Hz) Butterworth filter (as recommended in [48]). We then identified fixations using a spatial dispersion algorithm via the EYEMMV toolbox for MATLAB [49]. Fixations parameters were set to a minimum duration criterion of 100ms and spatial dispersion of 1° of visual angle (as recommended in [50]). The clubhead ositional data was filtered (10Hz lowpass Butterworth filter [51]) and initiation of the club head swing was automatically recognised based on changes in x-plane velocity of the Vive tracker, identified using peak detection in MATLAB. If the putter swing could not reliably be detected (e.g., if there was a lot of clubhead movement before and after the stroke) a trial was marked as 'missing' which is why some trials are missing for some participants. The final fixation beginning prior to this event, directed to the location of the ball (within 3° of visual angle) was selected as the quiet eye fixation. QE offset occurred when gaze deviated from the target (ball or fixation marker) by more than 3° of visual angle, for longer than 100ms [6, 38]. The absence of a QE fixation was scored as a zero. The location of gaze in the virtual environment could be calculated using a gaze vector from the known spatial

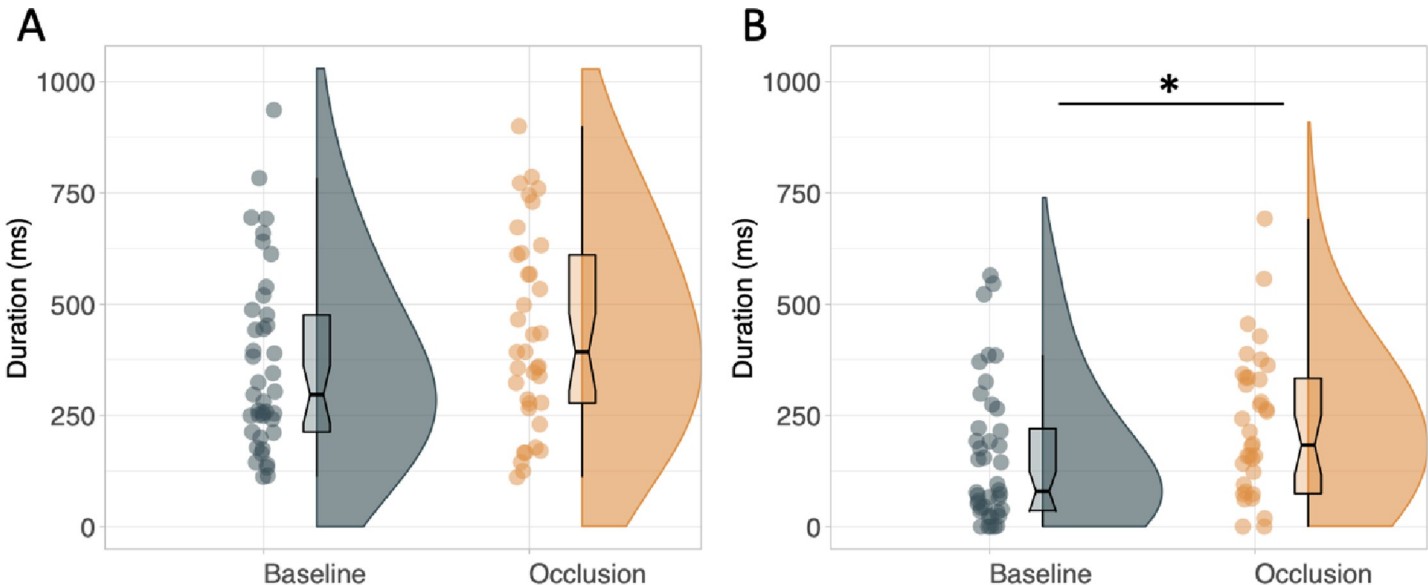

**Fig 2. Quiet eye results.** Raincloud plots with mean (black line) and median (boxplot notch) of total quiet eye durations (Panel A) and online portion of the quiet eye (Panel B). * $p$ = .05.

orientation of the head and the gaze in head direction. All analysis code is available from the Open Science Framework: https://osf.io/35fgp/.

Previous work has distinguished three distinct phases of the quiet eye fixation [6, 52, 53], corresponding to distinct phases of movement in this task, so we also adopted these definitions to further examine their functional role. The duration of the fixation occurring before the initiation of the movement (the start of the backswing in this case) is known as 'early quiet eye' and is proposed to reflecting a pre-programming function [52]. The period of the fixation that continues from the initiation of the swing until the moment of contact, known as 'online quiet eye', is thought to reflect online control of the action. Finally, the 'dwell' period of the quiet eye, which occurs when quiet eye continues after contact with the ball (or release in the case of a throwing task) can offer no functional benefit but may reflect good attentional control. Vickers [6] recommends that the quiet eye should ideally extend into a dwell period. To further examine the effect of the occlusion condition on quiet eye we also assessed the effect on these three separate periods of quiet eye.

**Putting performance.** Putting performance was assessed based on radial error of the ball from the hole, as is common in recent quiet eye and targeting tasks [10, 18, 19, 54]. The two-dimensional Euclidean distance (in cm) between the centre of the ball and the centre of the hole was measured automatically in the virtual environment. Performance was therefore a continuous measure of accuracy, with putts landing on top of the hole assigned an error of zero. An additional, exploratory, measure of performance was also calculated, but only for graphical purposes (see Figs 3B and 6B). A 95% confidence interval ellipse was calculated for each group based on the landing positions of individual putts. The 95% confidence ellipse defines the region that is likely to contain the mean of all future samples, based on an underlying Gaussian distribution.

**Putting kinematics.** Three putting kinematic variables were calculated to indicate the quality of the swing/impact [55–57]. The variability in stroke path was calculated by averaging the

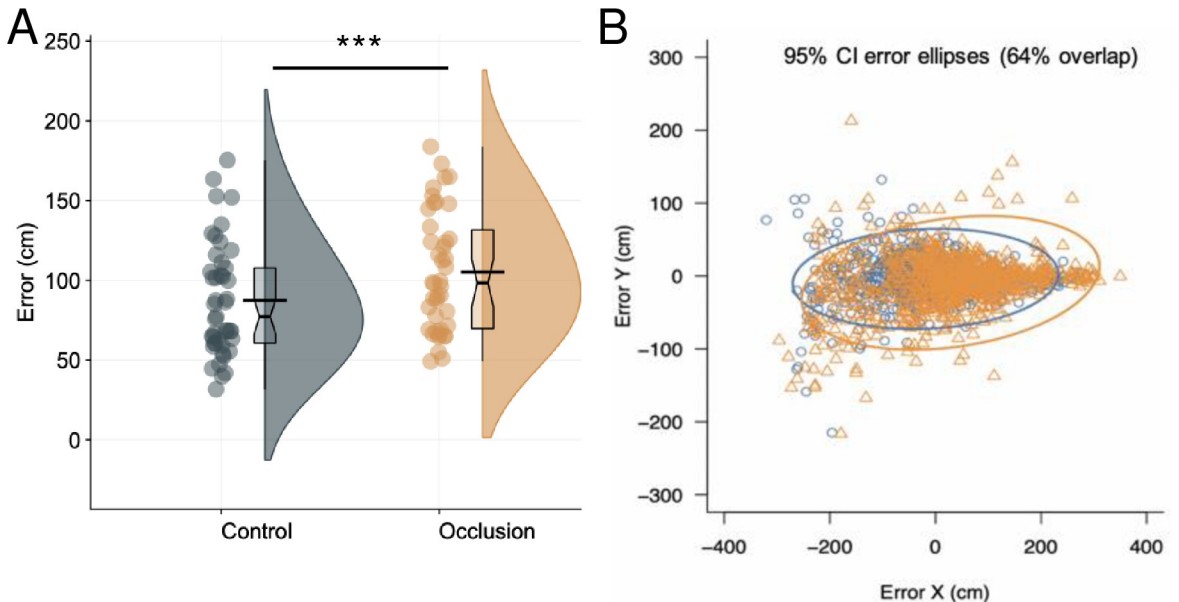

**Fig 3. Performance results.** Panel A: Raincloud plots, with mean (black line) and median (boxplot notch) of putting performance (radial errors) in control and occlusion conditions. Panel B: Error ellipses (with 95% CIs) for control (dark grey circles) and occlusion (orange [light grey] triangles) conditions. ***significant at $p < .001$.

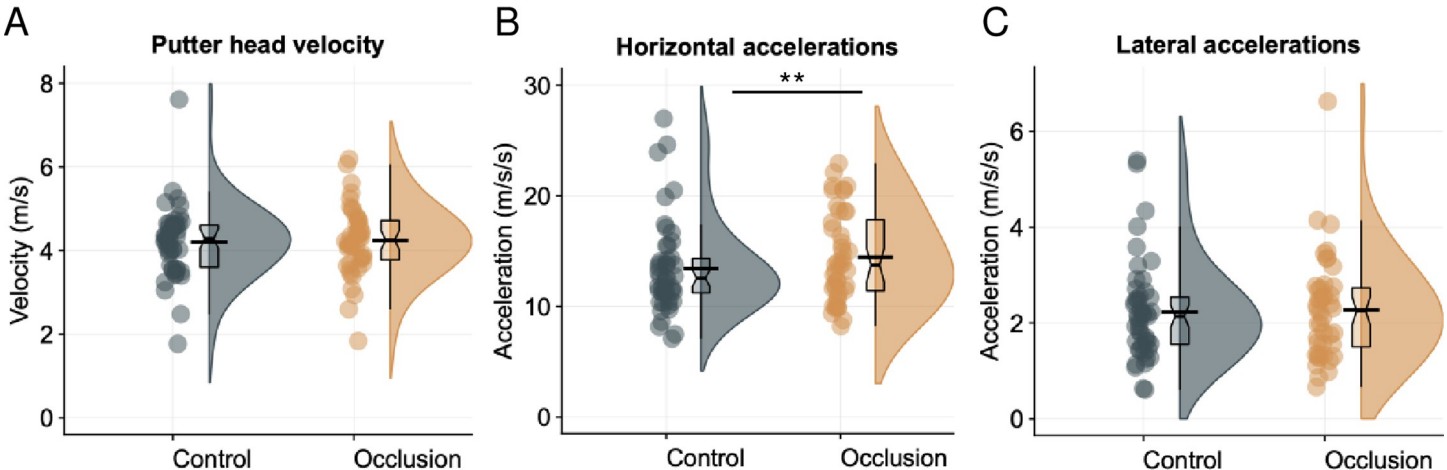

**Fig 4. Kinematics results.** Raincloud plots, with mean (black bar) and median (boxplot notch), of kinematic variables between conditions (putter head velocity at contact [A]; horixontal accelerations [B]; lateral accelerations [C]). **significant at *p* < .01.

accelerations (second derivative of position with respect to time) of the putter head during the downswing for the horizontal plane (x axis–the plane in line with the downswing), and the lateral plane (z-axis–the plane perpendicular to the downswing). The velocity (first derivative of position with respect to time) of the putter head at the moment of contact with the ball was also calculated. Kinematic data was de-noised using a five-point moving-average lowpass filter (10Hz).

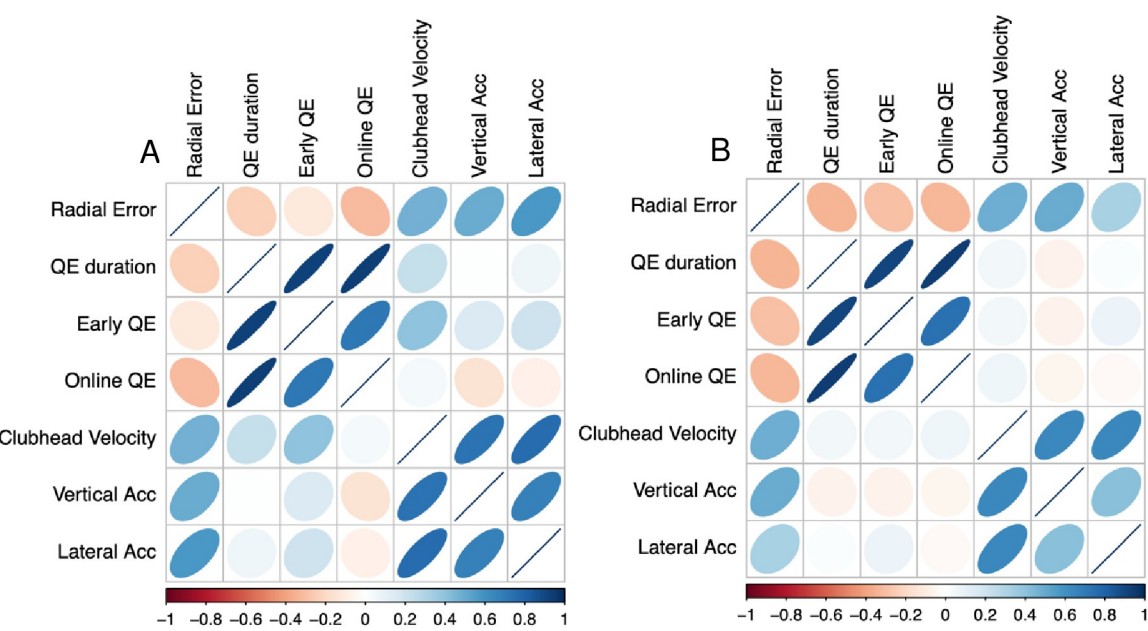

**Fig 5. Correlation heat plots.** Plots show correlation matrices for person level data for all dependent variables in control (Panel A) and occlusion conditions (Panel B). Colour and shape of the ellipse represent strength of the relationship. Panel A illustrates a weak correlation between longer quiet eye and reduced radial errors in the control putting condition. Panel B indicates that this correlation was not only maintained but was stronger in the visual occlusion condition.

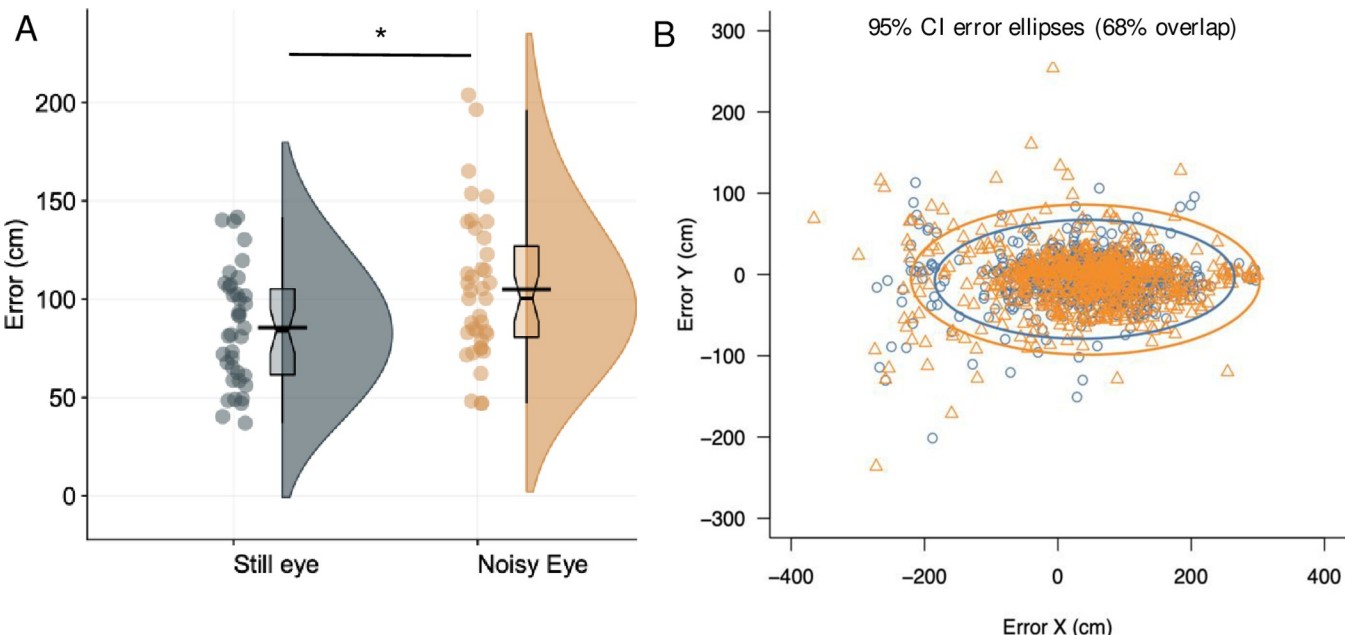

**Fig 6. Performance results.** Panel A: Raincloud plot with mean (black line) and median (boxplot notch) for putting performance. Panel B: Putting errors (with 95% CI ellipses) for still eye (grey circles) conditions and noisy eye (orange [light grey] triangles) conditions. *significant at $p < .05$.

## Procedure

Participants attended the lab for one visit lasting ∼45 minutes. After having the details of the experiment explained to them and giving written informed consent, participants were fitted with the VR headset. They then completed 3 familiarisation putts followed by 21 putts in each of the standard putting (control) condition and the occluded vision condition in a counterbalanced order. After each putt had been taken and the landing position recorded by the environment, the next trial was loaded and the ball returned to the starting position. Prior to each block of putts (i.e., each condition), the eye tracker was calibrated over 5 points in the visual scene. Participants were instructed to putt to the best of their ability and land the ball as close to the hole as possible. At the end of the study they were given the chance to ask questions and were debriefed.

## Data analysis

Data analysis was performed in RStudio v1.0.143 [58]. Data was checked for skewness and kurtosis and outlying values (more than 3 standard deviations from the mean; [59]) were identified. Outliers (individual trials) were replaced with a Winsorized score by changing the outlying value to a value 1% larger (or smaller) than the next most extreme score. A linear mixed effects model (LMM) was used to examine the effect of condition (free versus occluded) on the primary outcome variable (putting radial error) using the lme4 package for R [60]. LMMs were also used to compare quiet eye durations and putting kinematics (horizontal and lateral accelerations and putter head speed at contact). We treated quiet eye values of zero as missing cases for analyses, but include the same analyses with zeros included in the supplementary files; these analyses produced the same conclusions in all cases. A limitation of much quiet eye research (with some exceptions [37, 41]) is that it has used averaged scores, and correlations at a person level, rather than examining individual trials [61]. The use of LMMs in

the present work may enable a more sensitive approach that avoids averaging in this way. LMMs are also more effective at apportioning error variance when there are multiple measurements per participant (see [62]). In order to determine the best fitting model we initially fitted a near maximal model with random factors for participant (slope and intercept) and trial (intercept) [63]. Principal Components Analysis, using the RePsychLing package, was then used to identify overfitting and simplify the model, as recommended by Bates, Kliegl, Vasishth, and Baayen [64]. The best fitting model was chosen by simplifying the structure based on the number of principal components that contributed to explaining additional variance. The Akaike information criterion was also used to compare the fit of competing models. To enable more informed decisions about null effects we also calculated Bayes Factors for LMMs using the BayesFactor package [65] and Jeffreys-Zellner-Siow (JZS) prior. We report BF which represents the probability of the data under the alternative (i.e. values greater than one indicate how much more probable the alternative model is, given the data, while values less than one indicate a greater likelihood of the null model). Rules of thumb for interpreting standardised regression coefficients (std. beta) are not straightforward but Acock [66] suggests they can be interpreted in a similar way to *r*, such that $< .2$ is weak, $.2–.5$ is moderate, and $>.5$ is strong). We follow this convention when referring to effects in this paper. All analysis scripts and raw data are available from the Open Science Framework: https://osf.io/35fgp.

## Experiment 1 –Results

### Quiet eye

To assess the effect of condition on quiet eye (see Fig 2), a linear mixed effects model was run with participant as a random factor, allowing for varying intercepts and slopes. The model's total $R^2$ was .40 and marginal $R^2$ was .01. The model's intercept was at 18.57 (95% CI [16.86, 20.28]). Within this model, the effect of condition was statistically non-significant and small in magnitude (beta = 1.51, 95% CI [-0.08, 3.10], *p* = .06; std. beta = 0.18, BF = 0.61), indicating no difference in quiet eye duration between visual occlusion (mean = 462.2 ± 284.1ms) and control (mean = 390.0±263.2ms). A post-hoc analysis was run to examine whether quiet eye durations declined over trials in the occlusion condition, i.e. whether participants learned it was of no benefit. The LMM indicated that there was no decline in durations over trials (*p* = .97, $R^2$ = .006, BF = 0.00) suggesting that this did not happen.

To further examine the effect of condition on quiet eye, we ran an exploratory analysis (not part of the pre-registration) on the three separate quiet eye periods: early, online, and late. All three models used participant as a random factor (slopes and intercepts). For the early portion, the model's total $R^2$ was .28 and marginal $R^2$ was .001. The model's intercept was at 238.61 (95% CI [194.93, 282.28]). Within this model, the effect of condition was statistically non-significant and small in magnitude (beta = 20.25, 95% CI [-22.22, 62.71], *p* = .35; std. beta = 0.08, BF = 0.66), suggesting no difference between visual occlusion (mean = 250.6±147.5ms) and control (mean = 235.7±143.8ms) conditions.

The model predicting online quiet eye had a total $R^2$ of .31 and marginal $R^2$ of .01. The model's intercept, was at 8.96 (95% CI [7.01, 10.91]). Within this model, the effect of condition was statistically significant (beta = 1.95, 95% CI [0.11, 3.79], *p* = .04; std. beta = 0.20, BF = 79.87) but can be considered as small, (occlusion mean = 215.2±166.6ms; control mean = 154.0±158.1ms). Finally, the model predicting quiet eye dwell could not be adequately fit to the data, because the majority of values were zero, so is not reported. These results indicated that while the early portion of the quiet eye was unaffected by the occlusion, the online part may have actually been slightly lengthened in response to the occlusion.

## Performance

To assess the effect of condition on putting performance (see Fig 3), a linear mixed effects model was run on radial error scores, with participant as a random factor, allowing for varying intercepts and slopes. The overall model predicting error had a total explanatory power (conditional $R^2$) of .26, and marginal $R^2$ of .02. The model's intercept was at 0.85 (95% CI [0.76, 0.95]). Within this model the effect of condition was significant (beta = 0.21, 95% CI [0.11, 0.31], $p < .001$, BF = 43030.53), with radial error scores higher in the occlusion condition (mean = 105.7±37.7cm) than the control (mean = 83.9±32.1cm) (see Fig 3). The effect can be considered as moderate (std. beta = 0.29).

## Putting kinematics

To assess the effect of condition on the performance of the putting stroke, linear mixed effects models were run on putting kinematic variables (see Fig 4). The overall models (with participant as a random factor) predicting velocity of putter at contact and lateral accelerations were not significant, ($p$s > .75, std. betas < .062). The overall model predicting horizontal accelerations during the downswing was significant (beta = 0.10, 95% CI [0.035, 0.16], $p = .002$, BF = 6.54) but the effect can be considered as small (std. beta = 0.14). The model's intercept was at 3.56 (95% CI [3.43, 3.68]). Consequently there was limited effect of condition on putting kinematics (visual occlusion mean = 2.3±1.1m/s$^2$; control mean = 2.1±0.9m/s$^2$).

**Quiet eye and performance.** To examine the relationship between quiet eye and performance, using trial-by-trial data, we ran a linear mixed effects model to test the relationship between quiet eye durations and radial errors for each condition separately. For the control condition, the model (with random intercepts for participant) had a total $R^2$ of .19 and marginal $R^2$ of .01. The model's intercept was at 0.97 (95% CI [0.76, 1.18]). Within this model, the effect of QED was statistically non-significant and very small (beta = -0.006, 95% CI [-0.02, 0.003], $p = .20$; std. beta = -0.07, BF = 0.19). For the occlusion condition, the model (again with random intercepts for participant) had a total $R^2$ of .32 and marginal $R^2$ of .002. The model's intercept was at 1.09 (95% CI [0.88, 1.30]). Within this model, the effect of QED was statistically non-significant and very small (beta = -0.003, 95% CI [-0.01, 0.004], $p = .39$; std. beta = -0.04, BF = 0.15), indicating that there was no clear relationship between quiet eye and performance in either condition.

## Experiment 1 –Discussion

The purpose of experiment 1 was to determine whether individuals maintained a quiet eye fixation even when no visual information was available to inform movement planning or guide putter-ball contact. In line with our primary (pre-registered) hypothesis, the LMM indicated no change in quiet eye duration when visual information was occluded. A Bayesian analysis provided evidence in favour of the null, but only weakly so (BF = 0.61). This preservation of the quiet eye fixation is particularly striking when considering that the fixation was maintained within 3˚ of the ball, even though it was not visible. Directing fixations to locations where objects were previously located has been linked to re-activation of memory representations [27, 28, 67], hence the quiet eye could have been beneficial in this regard. An exploratory analysis actually revealed a significant increase in online quiet eye during the occlusion condition, from 120ms to 163ms. This was a small effect (std. beta = 0.16), but in the context of the limited duration of the swing–about 1000ms in novices [68]–this may be a meaningful increase. Under standard putting conditions the online portion of the quiet eye is often identified as supporting online control of movement and putter ball contact [53, 69]. Longer online quiet

eye in the current context might then reflect an increase in effort or attention to guide contact by recalling the spatial position of the ball (e.g., [70]).

Next, there was found to be an increase in putting error in the occlusion condition. This is an unsurprising result, given that all visual information was hidden during the putting movement. What is perhaps surprising is that the performance decrement was relatively small. Mean radial errors increased from 88cm to 106cm (std. beta = 0.29; see Fig 3), hence performance was not totally disrupted even when there was very little visual information to support the execution of the putt. The limited decrement suggests that much of the movement planning for a self-paced skill like the golf putt can be achieved with only a very brief preview of the target [35, 36], perhaps through using internal predictive models to guide the action [71]. Additionally, there was little evidence that the lack of visual information had a disruptive effect on control of the putting movement, as evidenced by the lack of changes in putting kinematics (see Fig 4)–although there was a significant increase in clubhead accelerations in the plane of the ball, the effect was very small. This suggests that the putting motion was not heavily reliant on online visual information and could be controlled in a more open-loop fashion.

Finally, while we predicted that a quiet eye-performance relationship would be maintained in the occlusion condition, we did not observe a relationship between quiet eye duration and radial error for *either* condition. Null within-subject effects have been reported in previous studies (see Lebeau et al. [4] for a summary) and may be related to the variance in performance arising from the novice sample. As no relationship was evident during either condition, it is inconclusive whether the maintenance of quiet eye during the occlusion condition provided a performance benefit. Correlation plots using averaged data (i.e. at the person, rather than trial, level; see Fig 5) did indicate a small to moderate negative correlation between putting error and quiet eye duration, in line with the expected relationship. This relationship was maintained in the occlusion condition (and actually appeared slightly stronger; Fig 5, Panel B), suggesting that this issue warrants further investigation.

Overall, these findings are not easily reconciled with the idea that the primary function of the quiet eye during the putting task was to gather visual information to plan, programme, and execute the putt (the outward-in role for QE). Visual occlusion had only a modest effect on putting accuracy, suggesting a very brief preview of the target was sufficient to enable much of the pre-planning. Additionally, as a quiet eye fixation was maintained even in the absence of visual information, the pre-shot fixation might serve a purpose beyond acquisition of visual information. However, as longer quiet eye durations were not clearly related to better performance in either condition, the findings are inconclusive as to whether maintaining the quiet eye was functional. Indeed, without evidence that the quiet eye duration in the occlusion condition was related to performance it is hard to discount that it could have just been maintained as part of a behavioural routine (although shot routines are less likely to be ingrained in our novice population). Therefore, in experiment 2 we further explored if there is a functional benefit to maintaining a stable fixation, outside of any visual processing role.

## Experiment 2 –Quiet eye v noisy eye

Experiment 2 aimed to directly test whether the stability of the quiet eye fixation has a functional benefit beyond acquisition of visual information. To do this we compared a stable point of gaze with a moving point of gaze, while equating visual input. As researchers have tended to assess quiet eye durations using manual coding procedures (where the gaze cursor must remain within one or three degrees of visual angle from the ball) rather than fixation detection algorithms [72], the quiet eye in most of the literature is not so much a fixation, but visual attention directed to an area. In addition, in some studies the definition of the quiet eye has

even been extended to moving locations [73], such that the quiet eye is no longer a fixation, but could be a smooth pursuit or tracking behaviour. Consequently, the importance of a single stable fixation remains unclear. Recent work by Gallicchio, Cooke and Ring [74] (see also [34]) using electrooculography found that although participants' eyes were most stable during the putting movement, there was no relationship between stillness of the eye and performance outcomes. Accordingly, a 'noisy eye' could still be functional.

As a development of the method used in experiment 1, we designed an experimental task which could induce a 'noisy eye' without also affecting visual input. In addition to the visual occlusion produced by the white sphere in experiment 1, in experiment 2 the ball and putter head were represented by red circles which enabled the participant to easily execute the putt by aligning the two red dots during the swing. A noisy eye was induced in one group by making the dots move, subtly, away from the participant at a speed of 1.5˚ visual angle per second (a distance equivalent to the radius of the ball) in the sagittal plane. As there was no reference background information it was ambiguous whether the dot was moving. The spatial relationship between the dots remained the same, such that they still accurately guided putter-ball contact. The dots began to move as soon as the occlusion began. Attending to the red dots representing the ball would now mean that participants could not maintain a stable fixation, but one that drifted across the visual field (i.e. not a true fixation, but a form of smooth pursuit). If the benefit of the quiet eye is indeed related to maintaining a stable fixation–the functional version of the inward-out perspective–then performance should be disrupted in the moving compared to the still condition. If, however, the quiet eye duration is just an emergent property of attentional control [15] or motor inhibition [75] the noisy fixation would not necessarily be disruptive to performance. Based on the findings of Gallicchio, Cooke and Ring [34], and an attentional control interpretation of quiet eye, we predicted that a noisy fixation would have no detrimental effect on putting accuracy.

## Experiment 2 –Methods

### Preregistration

The research question, hypotheses, sampling plan, methods, materials, and statistical analyses were all pre-registered on the Open Science Framework and can be accessed online (https://osf.io/ab6wu/). Any additional analyses not present in the preregistration are specified in the analyses as exploratory.

### Design

An independent groups design was used, with participants completing either the stable dots (still eye) or moving dots (noisy eye) condition. While it was unlikely that participants could recognise that the dots were moving in the 'noisy eye' condition, an independent design was chosen in case completing both conditions enabled participants to recognise the difference, which could influence their performance expectations. The primary outcome measure was putting accuracy (radial error in cm), and secondary measures were putting kinematics (accelerations in x,y,z) and quiet eye durations (in milliseconds), as in experiment 1.

### Participants

An entirely different sample of participants was recruited for experiment 2, again from the Undergraduate population. 80 participants (39 female), all novice golfers, volunteered to take part in the study. Sample size determination was based on detecting a 10cm change in radial error. A 10cm change was selected as it represented a small change in performance, based on

typical performance accuracy and variances in previous VR golf putting studies using novice performers (e.g., [76]). Monte Carlo simulations ($n = 1000$) indicated that 85% power was reached for a sample size of $\sim 75$. The plotted power curve is available from the supplementary materials (https://osf.io/56fmv/). Participants were provided with details of the study and gave written informed consent on the day of the testing visit. Ethical approval (REF: 191023-A-08) was obtained from the School of Sport Science Research Ethics Committee prior to data collection. The study, and collection of data, took place between November 2019 and March 2020. Authors had access to participants' identifying information during the data collection, but this was destroyed when the study was completed.

### Procedure

Other than the addition of stationary or moving dots to represent the ball and clubhead, all materials and measures were the same as in experiment 1. Participants attended the lab on one occasion for approximately 30 minutes and provided written informed consent at the start of the visit. They first completed 20 standard VR putts (visual information not occluded) to familiarise themselves with the putting simulator and were then randomly assigned to the still or noisy condition. Next they completed three practice and 20 trial putts in their assigned condition. Participants were instructed to putt to the best of their ability and land the ball as close to the hole as possible. Participants were finally debriefed and thanked for their participation. Data analysis procedures were as detailed in experiment 1.

## Experiment 2 –Results

### Quiet eye

As a manipulation check, to ensure that long stationary fixations were indeed disrupted by the manipulation, a mixed effects model was run on quiet eye durations. The model, with random intercepts for, had a total explanatory $R^2$ of .39 and marginal $R^2$ of .03. The model's intercept was at 449.96 (95% CI [336.24, 563.67]). Within this model, the effect of condition was statistically significant and moderate in size (beta = 168.84, 95% CI [9.72, 327.97], $p = .04$; Std. beta = 0.32, BF = $1.45*10^5$) indicating that the 'noisy eye' manipulation did indeed reduce the duration of quiet eye *fixations*, even though gaze may have remained on the location of the moving dot (nosiy eye mean = 287.9±229.4ms; still eye mean = 459.9±447.8ms).

### Performance

To assess the effect of the manipulation on the primary outcome, putting performance, a mixed effects model was run to compare the effect of condition on putting radial errors (see Fig 6). The model, with random factors for participant and trial number, had a total explanatory power (conditional $R^2$) of .27, in which the fixed effect of condition explained 1.53% of the variance. The model's intercept was at 0.95 (95% CI [0.89, 1.01]). Within this model the effect of condition was significant (beta = -0.081, 95% CI [-0.16, -0.0042], $p = .04$, BF = 318.07) and can be considered as moderate in magnitude (std. beta = -0.25), indicating that the 'noisy eye' did impair putting accuracy (nosiy eye mean = 105.1±0.37cm; still eye mean = 88.6 ±0.35cm).

### Putting kinematics

To examine whether putting condition affected motor skill execution, mixed effects models were run on putting kinematic variables (see Fig 7). The models predicting velocity at putter contact, lateral accelerations, and horizontal accelerations of the club head (with random

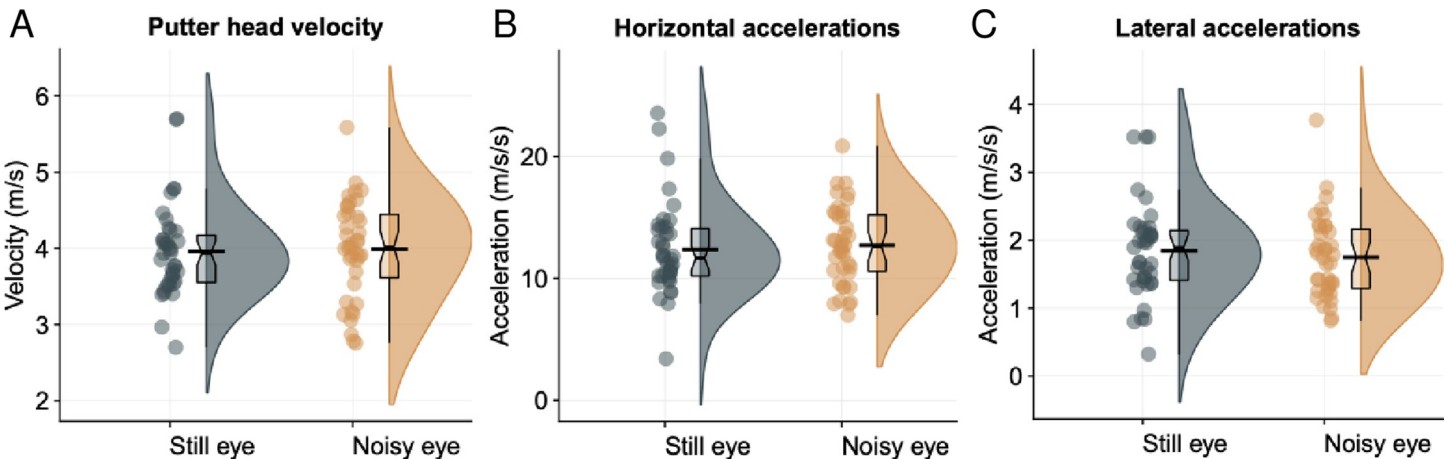

**Fig 7. Kinematics results.** Box plots with mean (black line), median (boxplot notch) for putting kinematics (putter velocity at contact [A]; horixontal accelerations [B]; lateral accelerations [C]).

intercepts for participant) were all non-significant with small effects ($ps > .31$; BFs $< 0.81$; std. betas $< 0.07$), indicating that the 'noisy eye' did not impact the kinematics of the putt.

## Experiment 2 –Discussion

Experiment 2 explored the importance of the stillness of the quiet eye by cueing still and 'noisy' pre-shot fixations during the golf putt. The manipulation check confirmed that quiet eye durations were indeed reduced in the noisy eye condition. Even though the reduction in the measured quiet eye durations was only a medium sized effect (std. beta = 0.31) from a mean of 460ms to 288ms, these remaining fixations were all moving at 1.5˚ visual angle per second and as such were not truly 'quiet'. In contrast to our pre-registered hypothesis, the induction of a noisy eye did have a detrimental effect on putting performance ($p = .04$, BF = 318.07). In addition, Fig 8 illustrates how correlations between performance and eye/ kinematic variables were disrupted in the noisy eye condition. This finding suggests that maintaining a still fixation (even in the absence of visual information) does provide some functional benefit. The impairment in performance would be conventionally described as small to moderate (std. beta = -0.25, equivalent to Cohen's $d = 0.46$), and in real terms equated to an increase in mean error from 89cm to 105cm. However, analyses of kinematic variables did not suggest any disruption of the motor execution of the putting stroke, as no differences were observed in velocity of the putter head at point of contact, or acceleration during the downswing in the horizontal (line of the swing) or lateral plane (perpendicular to the swing). That our measures of kinematics were not disrupted when performance was suggests that while these variables were related to performance (Fig 8), they did not fully capture the elements of the visuomotor response that are necessary for successful outcomes.

The disruption to performance resulting from the noisy eye condition contrasts with the findings of Gallicchio, Cooke and Ring [74], who reported no relationship between eye stillness and performance using electrooculography. The differing results might reflect the greater power of experimental manipulations in contrast to a purely correlational approach. In relation to wider questions about the functional role of the quiet eye fixation, the results from experiment 2 support a functional form of the inward-out perspective as a benefit of the quiet eye was demonstrated independently of any visual information acquisition. We discuss implications for theories of quiet eye function below.

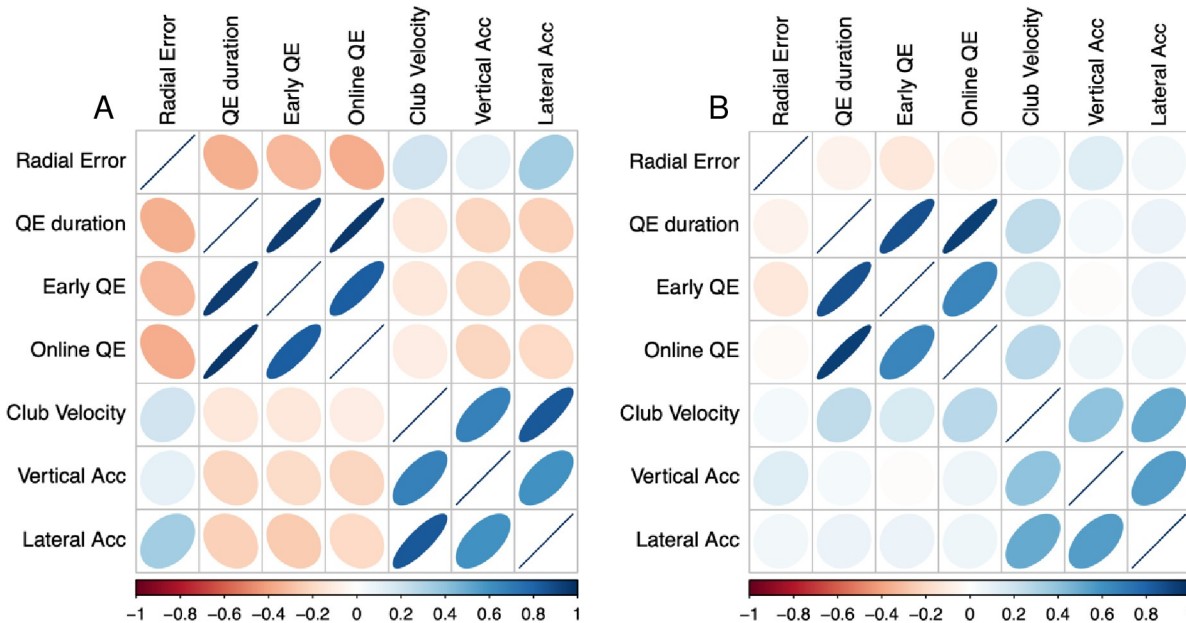

**Fig 8. Correlation heat plots.** Plot shows correlation matrices (person level data) for all dependent variables in still (Panel A) and noisy eye (Panel B) conditions. Colour and shape of the ellipse represent strength of the relationship. Weaker correlations across panel B suggest that the relationships between visual attention, motor control, and performance outcome were disrupted by inducing a 'noisy eye'.

## General discussion

The quiet eye has received considerable research interest as a route to understanding visual guidance in complex motor tasks. However, Williams [77] (p.116) suggests that "the greatest shortcoming in this area of study is the paucity of work that has attempted to better identify the mechanisms that underlie the quiet eye phenomenon." We have suggested that a particular issue is whether the quiet eye duration reflects an outward-in role–facilitating the acquisition of visual information to inform movement planning and execution–or whether it instead originates in an inward-out manner, as a reflection of ongoing internal processes. To begin to address this theoretical gap, the present study adopted novel manipulations of quiet eye made possible by virtual reality technology. The results have important theoretical implications for understanding the functional role of the pre-movement fixation in far-aiming tasks and the visual guidance of action more generally.

In experiment 1, the maintenance of the quiet eye duration in the absence of any relevant visual information (due to visual occlusion) suggested that the final pre-movement fixation may serve a purpose other than simply the acquisition of visual information for pre-programming the skill. However it is not yet clear what this purpose is. Previous work occluding portions of the quiet eye has also shown that participants persist in directing pre-shot fixations to where an occluded target would be [52]. Taken together, these findings indicate that, in a self-paced task, the quiet eye duration may be taking on an inward-out role, such as assisting and supporting spatial memory, attentional control, inhibition of movement alternatives, and/or neural quiescence. In experiment 1 we could not, however, conclusively determine whether maintenance of a longer quiet eye during occlusion conditions served a *functional* inward-out role, or was just part of a behavioural routine. Experiment 2 subsequently confirmed that maintaining a still fixation was indeed functional for performance, regardless of visual information, which might in turn explain the preservation of the quiet eye during experiment 1.

## Quiet eye mechanisms

It has previously been demonstrated that manipulating parameters of the quiet eye, such as the duration (long versus short) and location (on the ball, above/beside the ball, or to the hole) has very little effect on the performance accuracy of novices in a virtual golf putting task [19]. When considered alongside the present findings, these results suggest that in the self-paced task of golf putting the quiet eye is not primarily performing an outward-in role. Instead much of the necessary visual information is likely acquired during initial previews of the ball and hole, and maintaining some kind of still fixation–on the ball or close to the ball [19], or even on nothing (Experiment 1&2 here)–may be sufficient to execute the skill. As discussed, this still fixation could be beneficial for a range of inward-out reasons, related to goal-directed attentional control and improved neural efficiency and the present findings appear to be supportive of a functional inward-out role, rather than just being an epiphenomenon of ongoing processing.

One potential explanation for the observed functional benefit of a still pre-shot fixation is a role in spatial memory. Spontaneous eye movements to blank locations that were previously associated with particular information have been shown to facilitate memory retrieval [28]. Fixating the ball might help maintain a representation of the ball's spatial position relative to the performer, even if no additional visual information is required or available. Alternatively, maintaining a still fixation has been proposed as a strategy to actively avoid processing unnecessary and disruptive external information when performing purely internal tasks [25, 30]. This suggestion is similar to the inhibition hypothesis of quiet eye [13, 75] which proposes that by adopting a stabilizing gaze on one task-relevant cue, movements can be more effectively synchronised and one movement plan can be selected, while alternatives are inhibited. This also aligns with prior suggestions that the quiet eye is linked to neural quiescence, a feature of expert-like performance [38, 78]. These explanations all provide viable explanations of how quiet eye might take on an inward-out role, but still provide a functional benefit.

It has previously been suggested that the function of the quiet eye likely varies across tasks [77], but long pre-shot fixations seem to be common. The balance between inward-out and outward-in determinants of the fixation duration may be one way of understanding this variation. In externally-paced and interceptive tasks longer fixations may be primarily driven by an outward-in role. But in self-paced tasks, a long fixation may be retained because of inward-out influences where a still fixation is still beneficial, but for other reasons. Skills like golf and soccer penalty taking, which involve acting on a near target to project it towards a far target, might lie in between, and both inward-out and outward-in factors may play a part; initially a quiet eye fixation could be driven by outward-in influences, but be maintained to support neural quiescence or attentional control. If quiet eye tasks can indeed be characterized along of a spectrum of functionality, the interesting questions are to determine which instances of the quiet eye are serving which function.

## Implications

Clearly further work is needed to develop the proposed characterisation of outward-in and inward-out influences on pre-shot gaze behaviours, but this dichotomy might offer several benefits. First, it may help to explain the expertise paradox [79], where skilled performers maintain a longer quiet eye despite apparently having a reduced need for increased processing time. Second, it might also be relevant for why longer quiet eye is not always better; if the fixation does not just serve to better process the scene, once the necessary information has been acquired and performers have appropriately prepared for the action (e.g., attentional control,

inhibition of alternative movements variants, or neural quiescence) a longer fixation may not provide additional benefits.

## Limitations

A limitation of all quiet eye research is the variation in approaches to calculating the quiet eye duration, which is partly due to variations in the eye tracking hardware and software employed. Many studies have used manual coding procedures where researchers determine the onset and offset of the fixation by watching the point-of-gaze videos and inferring when the gaze is on or off the target [2, 80]. When the positions of objects in the scence and the participant's head can be tracked, automated approaches can be used to determine when gaze is on the target [19, 21]. This approach reduces the subjectivity of determining whether the gaze remains steadily on the target for long enough to count. Some eye trackers also come with software that employs fixation detection algorithms to determine gaze events based on spatial dispersion or velocity of eye position [50]. This improves the subjectivity of determining when a fixation event has occurred, but the resultant quiet eye durations are still dependent on the criteria chosen for defining a fixation (e.g., degrees of dispersion or minimum durations). As a result, the exact durations reported in studies is very much dependent on the data collection and analysis choices made in the study. We see the algorithmic approach to fixation detection used in the present study as a strength because of the more objective and replicable methodology. The approach we used does mean, however, that the fixation criteria were stricter than many previous studies, because trials on which a clear fixation on the ball could not reliably be found were treated as a zero duration quiet eye. As a result the quiet eye durations we reported were shorter (or more frequently absent) and therefore perhaps not directly comparable to many previous studies.

A strength of the current work is the occlusion manipulations that were made possible by the use of virtual reality, as similar physical set-ups would have been challenging to implement. While this was a visually guided motor skill which captured key aspects of putting, the degree to which this virtual task remained representative of real putting is still to be determined [81, 82]. Another important consideration when interpreting the present results is the novice population. The quiet eye has been examined across both novices and experts in a range of tasks, but the relative importance of the pre-shot fixation may vary between skilled and unskilled participants [4]. For instance, experts may have a better ability to maintain control of their visual attention, and may be better able to refine their well-developed motor models [83] during the pre-shot preparation period. Consequently, the exact functionality of the quiet eye may be moderated by experience, much as it is by task. Hence, we should be cautious about extending these findings too far beyond the current task (which was clearly different from real putting) and population.

## Conclusion

Few specific variables in the study of visual guidance of action have attracted as much interest as the quiet eye, yet we still do not know exactly why a longer fixation to the target provides benefits for action. While previous research has shown that training longer quiet eye durations can improve performance [84, 85] it is not clear how this relationship is driven and better explanations may require measures that target underpinning cognitive processes and not just the specifics of the fixation. The present findings indicate that performers will maintain a quiet eye even in the absence of visual information, suggestive of inward-out drivers. Additionally, a still fixation appears to be beneficial irrespective of visual input, which suggests that the inward-out role may also be functional. These are novel findings which contribute to a new

theoretical understanding of quiet eye and raise a number of important questions for further work to clarify the exact functional role of stabilising gaze during visually-guided behaviour.

## Acknowledgments

The authors would like to thank Connor Burns, Josh O'Hara, Megan Walters, and Eden Theodore for help with data collection.

## Author Contributions

**Conceptualization:** David J. Harris, Mark R. Wilson, Samuel J. Vine.

**Data curation:** David J. Harris.

**Formal analysis:** David J. Harris.

**Methodology:** David J. Harris, Mark R. Wilson, Samuel J. Vine.

**Supervision:** Mark R. Wilson, Samuel J. Vine.

**Visualization:** David J. Harris.

**Writing – original draft:** David J. Harris, Mark R. Wilson, Samuel J. Vine.

**Writing – review & editing:** Mark R. Wilson, Samuel J. Vine.

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
