## [Decision Letter · Decision Letter 0]

15 Jun 2023

PONE-D-23-08839The functional role of visual information and fixation stillness in the quiet eyePLOS ONE

Dear Dr. Harris,

Thank you for submitting your manuscript to PLOS ONE. After careful consideration, we feel that it has merit but does not fully meet PLOS ONE’s publication criteria as it currently stands. Therefore, we invite you to submit a revised version of the manuscript that addresses the points raised during the review process.

We look forward to receiving your revised manuscript.

Kind regards,

Nick Fogt

Academic Editor

PLOS ONE

“DH’s time was supported by a Royal Academy of Engineering UKIC Fellowship”

4. We note that Figure 1 in your submission contain copyrighted images. All PLOS content is published under the Creative Commons Attribution License (CC BY 4.0), which means that the manuscript, images, and Supporting Information files will be freely available online, and any third party is permitted to access, download, copy, distribute, and use these materials in any way, even commercially, with proper attribution. For more information, see our copyright guidelines: http://journals.plos.org/plosone/s/licenses-and-copyright.

b.If you are unable to obtain permission from the original copyright holder to publish these figures under the CC BY 4.0 license or if the copyright holder’s requirements are incompatible with the CC BY 4.0 license, please either i) remove the figure or ii) supply a replacement figure that complies with the CC BY 4.0 license. Please check copyright information on all replacement figures and update the figure caption with source information. If applicable, please specify in the figure caption text when a figure is similar but not identical to the original image and is therefore for illustrative purposes only.

Additional Editor Comments:

Both reviewers had positive things to say about the manuscript. However, there are some concerns that will need to be addressed. Reviewer #1 raises the point that accuracy improved in spite of a lack of change in kinematics. This could suggest that the kinematic measures do not apply to the actual task. In addition, please provide more information around how the quiet eye was defined in terms of threshold values for fixation.

Reviewer #2 raises a number of concerns, the most significant of which is that there are appear to be a large number of trials with no quiet eye periods. The definition of "missing trials" is unclear perhaps as a result. Please clarify these points, and provide a guide as to how to process/read the files.

Reviewers' comments:

Reviewer's Responses to Questions

**Comments to the Author**

1. Is the manuscript technically sound, and do the data support the conclusions?

Reviewer #1: Yes

Reviewer #2: Yes

2. Has the statistical analysis been performed appropriately and rigorously? 

Reviewer #1: Yes

Reviewer #2: Yes

3. Have the authors made all data underlying the findings in their manuscript fully available?

Reviewer #1: Yes

Reviewer #2: Yes

4. Is the manuscript presented in an intelligible fashion and written in standard English?

Reviewer #1: Yes

Reviewer #2: Yes

5. Review Comments to the Author

Reviewer #1: Very nice experimental study, and nice work using VR to manipulate the variables. High internal validity-not sure about the ecological validity though. The authors stated this limitation in the discussion section, but I am wondering how those results would extend to an expert population in a real golf putting setting.

Good work stating the problem and summarizing the mechanisms in the introduction. I would suggest adding subtitles in this section to better guide the reader.

Line 166-167: “The study was powered to find the smallest meaningful effect of interest,

which was set at a 75ms change in quiet eye duration.” Can the authors expand on why this 75ms change was selected? Same for experiment 2, why 10cm and not 15 or 5?

What commitment/effort measured? Did participant have an incentive to perform well? Again, this relates to the ecological validity of the study.

Good work discussing the results of experiment 1.

How can we explain that the putting accuracy was affected but not the putting kinematics in experiment 2? Does that mean the variables selected for the putting kinematics were not predictive of performance?

Discussion is well written, summarizes the results well, and good comparison/contrast with previous studies. This is a very good paper that advances the literature in the mechanisms underlying the QE.

I am glad the authors mentioned task (i.e., self-paced vs externally-paced) as a moderator. This is indeed an important variable, as is expertise level.

Same as introduction, the discussion would benefit from having subtitles.

Minor:

Check APA style for in-text references and reference list.

Typo line 354

Reviewer #2: I appreciate the chance to review this manuscript. Indeed, the authors' proposal of the inward-out versus outward-in role of fixation can bring a new perspective to the functional role of the quiet eye. Concerning the manuscript's data, however, there is a significant concern. When examining OSF data, the available files are difficult to comprehend and interpret, and there are numerous instances in which participants lack fixation. In comparison to existing literature, these instances occurred far more frequently. I will inquire further to clarify this point, but as of now, the main conclusion that fixation is not necessarily relevant to information processing is not sufficiently substantiated.

P 2 L 30-31

It is not typical to mention effect size in the introduction. I recommend that the author keep their original idea but eliminate the effect sizes and confidence intervals.

P 4 L 94-95

Since the empirical evidence for the differences between experts and novices is consistently supported by the literature (e.g., Lebeau et al., 2016), it cannot be concluded that athletes do not require longer fixations simply because there could be an alternative explanation for the quiet eye, as you mentioned, for instance because they could be focused and attentive. Whether they truly need it or not is a conclusion that follows directly from the empirical evidence we have gathered thus far.

P5 L 114

Different studies, contrary to Horn and Marchetto, found that a higher degree of certainty regarding the upcoming target location was associated with the preprogramming function of the quiet eye. For example, see Vincze et al., (2022). Quiet eye facilitates processing complex information in elite table tennis players. Visual Cognition, 30(7), 506-516. I recommend that authors include additional positions and emphasise the significance of distinguishing between the different roles of the QE.

P 9 L 235

The link to the analysis code is not uploaded accurately. The power curves are opened when attempting to open the link from P 9, l 235.

P9 L236-244

Can the authors articulate on the "three distinct phases" and how they pertain to sport performance? Given the fact that a singular fixation is ‘artificially’ divided into three phases, does this provide additional insight into the phenomenon?

P10 L 288

When examining the OSF files, there are a number of participants with more than fifty percent of trials with 0 Quiet Eye.

Can the authors initially offer a potential explanation for why this occurred?

I find this result intriguing, since I have not identified more than 9% of trials without a final fixation in the literature.

On the pre-registration form (at point 23), the authors specified that a participant would be eliminated if 50 percent of trials were missed. Those trials performed without a final fixation are not regarded as 'missing'? Then, how did the authors define their 'missing trials'? Moreover, there are quite large variations in the total number of trials performed by participants. Can the authors clarify why this is taking place?

Can the authors provide raw data for a participant, such as participant 24PS, who has extensive trials with 0 Quiet Eye?

I would strongly suggest that they include a legend in their OSF file so that they can easily identify which data is from the first experiment, which is from the second, how many trials represent the occluded condition, etc. It is currently very difficult to navigate through their files.

P10 L 290

I commend the authors for mentioning this limitation, as it is both true and unfortunate that the majority of the literature relies on averaged scores. However, there have been previous attempts to address this limitation, so I recommend that the authors highlight these efforts. See, for example, Vincze et al., 2022, cited above.

P 17 L 460

For a better understanding of the, a visual representation of the moving dots would be of great assistance.

P 19 L497

Can the authors additionally provide a number/code for their ethical approval and include it in the manuscript? I would also request that they submit the ethical approval when addressing the review procedure.

P23 L 621

Can the authors discuss a potential neurological explanation for the inward-outward phenomenon?

P25 L 675

Can the authors address any limitations of the employed task or procedure? The limitation mentioned is quite general and applicable to a number of studies investigating Quiet Eye.

General note

I have a limited comprehension and application of Bayes factor and it's possible that my evaluation skills on this subject are poor. But, based on what I know, Bayes is fairly taxing when p is close to .05. I find the 'noisy eye' results (p =.04, BF = 318.07) to be somewhat inconsistent or whether there is an error in reporting. In addition, when considering the other results presented in the manuscript, ps and Bs "are essentially moving in the same direction." Could the authors explain how and why is this possible in this case?

General (minor) note

Perhaps this is just a personal preference, but I find that five citations for a single argument are quite a lot. In addition, certain sources are improperly cited in the manuscript (see for example P 8, l 217, l 220).

6. PLOS authors have the option to publish the peer review history of their article (what does this mean?). If published, this will include your full peer review and any attached files.

Reviewer #1: No

Reviewer #2: No

---

## [Author Response · Author response to Decision Letter 0]

30 Jun 2023

Responses to reviewer comments

Response: We want to thank the two expert reviewers for taking time to appraise our work. We have considered all the comments and provided a point-by-point response below. We hope the responses and changes to the manuscript address any concerns they have. 

Comments from the editor

Response: Checked, and formatting changes made. Thanks. 

“DH’s time was supported by a Royal Academy of Engineering UKIC Fellowship”

Response: Amended.

Response: Name of the board and approval number added. We do already state that the participants gave written consent. 

4. We note that Figure 1 in your submission contain copyrighted images. All PLOS content is published under the Creative Commons Attribution License (CC BY 4.0), which means that the manuscript, images, and Supporting Information files will be freely available online, and any third party is permitted to access, download, copy, distribute, and use these materials in any way, even commercially, with proper attribution. For more information, see our copyright guidelines: http://journals.plos.org/plosone/s/licenses-and-copyright.

Response: Thanks for checking, but these aren’t copyrighted. The image on the left is of one of the participants who gave consent for the photo and the picture of the software is software that I wrote. So pretty sure there are no issues. 

Comments to the Author

Reviewer #1: Very nice experimental study, and nice work using VR to manipulate the variables. High internal validity-not sure about the ecological validity though. The authors stated this limitation in the discussion section, but I am wondering how those results would extend to an expert population in a real golf putting setting.

Response: Thanks for your positive comments. Yes, we fully agree that extending these findings to expert golfers in the real-world is difficult and that further work would be needed to be able to make any strong claims. The aim of the current work was really to look at some of the underlying perception-action mechanisms. We would argue that these mechanisms should be similar in experts and therefore we can learn something from novices still, and it enables a much larger sample than typically used in this research area. But we do note the limitations of our approach at lines 663-670

Good work stating the problem and summarizing the mechanisms in the introduction. I would suggest adding subtitles in this section to better guide the reader.

Response: Thanks, added as suggested. 

Line 166-167: “The study was powered to find the smallest meaningful effect of interest,

which was set at a 75ms change in quiet eye duration.” Can the authors expand on why this 75ms change was selected? Same for experiment 2, why 10cm and not 15 or 5?

Response: Thanks, yes the values we chose are somewhat arbitrary, but reflect what was considered to be a small change, based on previous literature. For example, a 75ms change in QE duration is pretty tiny in the context of the amount of variance in these values (e.g., less than the shortest possible fixation), so power to detect anything down to this value seemed a good rationale for powering the study. But yes, this could equally have been 70ms or 80ms. We have added some additional description of this to the paper for both experiment 1 and 2:

“The study was powered to find the smallest meaningful effect of interest, which was set at a 75ms change in quiet eye duration. The 75ms value, while slightly arbitrary, was selected as a change this small would be considered very small in relation to previous literature and typical variances in QE values among novices.”

“Sample size determination was based on detecting a 10cm change in radial error. A 10cm change was selected as it represented a small change in performance, based on typical performance accuracy and variances in previous VR golf putting studies using novice performers (e.g., [75]).”

What commitment/effort measured? Did participant have an incentive to perform well? Again, this relates to the ecological validity of the study.

Response: No there was monetary incentive or anything but we find that people generally want to perform well in any sporting skill. Especially a population of predominantly sport science students. So we don’t have any data to confirm it, but anecdotally people were pretty engaged in the task and doing their best. Also, our focus wasn’t on achieving high ecological validity – we don’t really claim that this task was a close replication of real golf (particularly competitive golf), it was instead an aiming task that allowed us to examine the quiet eye in a controlled way. 

Good work discussing the results of experiment 1.

Response: Thanks. 

How can we explain that the putting accuracy was affected but not the putting kinematics in experiment 2? Does that mean the variables selected for the putting kinematics were not predictive of performance?

Response: Presumably kinematics could still be predictive of performance but not affected by the manipulation. Indeed, as noted in footnote 5, these measures were correlated with performance in both conditions. 

However, as you say, these findings effectively show that these variables are not fully capturing the important aspect of performance if outcome can be disrupted without measurable changes in the kinematics. We have added this to the text (lines 570-572). 

The potentially more interesting issue is the (lack of) connection between the visual information and the putting motion in experiment 1– that these variables did not change in the absence of vision suggests that they were not primarily controlled by online mechanisms but by a more open loop form of control that does not rely on vision. Or at least could be controlled in this way. We have added another sentence on this (line 402). 

Discussion is well written, summarizes the results well, and good comparison/contrast with previous studies. This is a very good paper that advances the literature in the mechanisms underlying the QE.

I am glad the authors mentioned task (i.e., self-paced vs externally-paced) as a moderator. This is indeed an important variable, as is expertise level.

Same as introduction, the discussion would benefit from having subtitles.

Response: Added, thanks for the suggestion. 

Minor:

Check APA style for in-text references and reference list.

Response: In response to the editors request we have changed to a numbered referencing style. 

Typo line 354

Response: Fixed, thank you. 

Reviewer #2: I appreciate the chance to review this manuscript. Indeed, the authors' proposal of the inward-out versus outward-in role of fixation can bring a new perspective to the functional role of the quiet eye. Concerning the manuscript's data, however, there is a significant concern. When examining OSF data, the available files are difficult to comprehend and interpret, and there are numerous instances in which participants lack fixation. In comparison to existing literature, these instances occurred far more frequently. I will inquire further to clarify this point, but as of now, the main conclusion that fixation is not necessarily relevant to information processing is not sufficiently substantiated.

Response: Thanks for the positive appraisal of the work. In relation to the ‘zero quiet eye’ issue we respond to the comment below. 

P 2 L 30-31. It is not typical to mention effect size in the introduction. I recommend that the author keep their original idea but eliminate the effect sizes and confidence intervals.

Response: Ok thanks. Removed. 

P 4 L 94-95. Since the empirical evidence for the differences between experts and novices is consistently supported by the literature (e.g., Lebeau et al., 2016), it cannot be concluded that athletes do not require longer fixations simply because there could be an alternative explanation for the quiet eye, as you mentioned, for instance because they could be focused and attentive. Whether they truly need it or not is a conclusion that follows directly from the empirical evidence we have gathered thus far.

Response: If we have understood your comment correctly (apologies if we haven’t), you are suggesting that because experts have consistently been found to have longer durations they do in fact need a longer quiet eye. 

We would argue that this is not necessarily the case as correlation does not equal causation. There is no empirical doubt that on average and across groups experts have a longer QE than novices… but there is little evidence to suggest that this is the cause of good performance, and even less evidence to suggest what exactly is happening during this period of time (i.e., the mechanism). There are also plenty of instances in individual data sets in which experts do not need a long quiet eye in order to maintain performance.

As has been articulated in papers on the ‘efficiency paradox’ (e.g., Klosterman & Hossner, 2018), if the QE is a time for planning the upcoming shot experts, who have more automated motor programmes, should actually find this easier and therefore have a reduced need for the QE. We are not saying this is necessarily true, or that experts never need a QE, but rather that in some instances it is possible that an expert might maintain a long QE despite it not really being necessary for performance (e.g., they can maintain good attention without their eyes being still). 

P5 L 114. Different studies, contrary to Horn and Marchetto, found that a higher degree of certainty regarding the upcoming target location was associated with the preprogramming function of the quiet eye. For example, see Vincze et al., (2022). Quiet eye facilitates processing complex information in elite table tennis players. Visual Cognition, 30(7), 506-516. I recommend that authors include additional positions and emphasise the significance of distinguishing between the different roles of the QE.

Response: Thanks, this is an interesting paper. However, we interpret the experimental conditions as rather different. In the Horn and Marchetto paper, participants were given additional early information about where the target would be (so they could start planning a bit earlier) which reduced the need for an early QE, showing that the visual information gained by the early QE wasn’t really that important when the location was known. But in the Vincze et al. paper it sounds like there was no manipulation of prior knowledge about where the target would be (as that was known in both conditions), it was just a much harder target to hit. This is more like previous studies that have manipulated task demands and found longer QE durations (e.g., Williams, Singer, and Frehlich, 2002; Walters-Symons et al., 2018). There are also other important differences, like the Vincze et al paper was an externally paced sport where the QE was a tracking gaze, rather than a self-paced aiming task. The point that we were trying to make in the paragraph you refer to is that while the QE might be important for things like locating a moving target – e.g. in tracking tasks – in simpler self-paced tasks it might not be. So the different results for these studies only really supports our point. However, we take the point that the Vincze et al study does support a response-programming explanation and have added it as a reference. 

P 9 L 235. The link to the analysis code is not uploaded accurately. The power curves are opened when attempting to open the link from P 9, l 235.

Response: Apologies. We have changed the link which should now go to the main project page. Thanks for spotting. 

P9 L236-244. Can the authors articulate on the "three distinct phases" and how they pertain to sport performance? Given the fact that a singular fixation is ‘artificially’ divided into three phases, does this provide additional insight into the phenomenon?

Response: There is logic in dividing the QE into these distinct phases in the skill of putting. The pre quiet eye is a part of the fixation that happens prior to any action, the online part happens during action, and the dwell occurs at a period when physical action can no longer influence the physics of the ball (and therefore outcome). Pre-programming and online control parts of the QE map nicely onto well discussed processes relating to the control of movement (e.g. predictive vs prospective control). Vickers provides a clear, albeit limited, description of what dwell might represent in terms of attentional allocation. 

It is really beyond the scope of this paper to discuss whether it is correct to divide the fixation into three phases, but as previous literature has split the QE into these distinct phases (with putatively different functions) we thought it could be useful to examine each of these separately (text added to the manuscript: line 229-230). Our data doesn’t allow us to determine whether this splitting up is valid or not so we can’t really speak to this issue. We are not aware of any papers that have specifically shown that one part or another is more predictive of performance outcomes but this would be interesting work to do. Although there may be functional relevance to the difference phases – e.g., breakdowns under pressure seem to be related more to the online portion (Vine et al. 2013). 

Vine, S. J., Lee, D., Moore, L. J., & Wilson, M. R. (2013). Quiet eye and choking: Online control breaks down at the point of performance failure. Medicine & Science in Sports & Exercise, 45(10), 1988-1994.

P10 L 288. When examining the OSF files, there are a number of participants with more than fifty percent of trials with 0 Quiet Eye.

Can the authors initially offer a potential explanation for why this occurred?

I find this result intriguing, since I have not identified more than 9% of trials without a final fixation in the literature.

Response: Thanks, yes this is an interesting issue. The method we used to calculate the quiet eye was based on an automated fixation detection method which is quite strict for a couple of reasons. Firstly, we used a fixation detection algorithm to identify periods of stable gaze (spatial dispersion of 1° of visual angle), whereas the vast majority of previous work has used manual coding where the cursor in a video has to remain within a certain region to be classed as a QE. The fixation detection approach is much stricter as any fixation that was not still enough would not be classified. So previous manually coded ‘fixations’ are not fixations in the truest sense and therefore give longer QE values. Secondly, we didn’t want to accidentally identify fixations that weren’t true fixations as a QE, and so we used a conservative method that only classified a QE if a clear fixation on the ball at the relevant time was found. As a result, there were a lot of trials that did not meet this criteria. Particularly with the novice population who would tend to have more variable gaze behaviour. So we understand that this is a little different from many previous studies, but the durations of the QE that you get depends on the definition you use (e.g., those using 3° have longer durations that those using 1°) and we essentially have a very strict definition. We do mention this issue in the limitations section of the paper where we acknowledge that the duration in this study may not be directly comparable to previous studies: “We see an additional strength in the use of the algorithmic approach to fixation detection which ensures a more objective classification than manual coding. This approach does mean, however, that the fixation criteria were stricter than many previous studies so the quiet eye durations we detected were shorter (or more frequently absent) and therefore perhaps not directly comparable to previous studies.”

On the pre-registration form (at point 23), the authors specified that a participant would be eliminated if 50 percent of trials were missed. Those trials performed without a final fixation are not regarded as 'missing'? Then, how did the authors define their 'missing trials'? 

Response: Missing trials were ones where there was either a data recording error or the phases of the trial could not be identified (i.e., the MATLAB code could not detect when the putter swing happened). We did not consider the participant not using a QE as an error, we consider it as a legitimate behaviour that we would argue should be captured and not thrown out. Also there was other data captured on those trials like performance and kinematics. As you will have seen, we ran the analysis both with zeroes treated as missing and also again (in the supplementary files) with zeroes included, which made no difference to the results. 

Moreover, there are quite large variations in the total number of trials performed by participants. Can the authors clarify why this is taking place?

Response: Yes, this was to do with the issue of identifying the phases of the swing reliably with the MATLB code. We used the movement of the putter to determine when the swing happened, but if for instance the participant was waving the putter around a lot and the code could not clearly detect when the swing started then the trial was discarded. So some people who had moved around a lot before and after a putt ended up with quite a few missing trials. So it wasn’t variation in the number of trials that they performed, just the number that we have measurements for. In subsequent studies we have given instructions for participants to not do ‘practice putts’ in between trials to reduce this data loss. We have also added the following text to the methods to explain this issue: “If the putter swing could not reliably be detected (e.g., if there was a lot of clubhead movement before and after the stroke) a trial was marked as ‘missing’ which is why some trials are missing for some participants.”

Can the authors provide raw data for a participant, such as participant 24PS, who has extensive trials with 0 Quiet Eye?

Response: Yes, we have added some examples for each study to the OSF repository. 

I would strongly suggest that they include a legend in their OSF file so that they can easily identify which data is from the first experiment, which is from the second, how many trials represent the occluded condition, etc. It is currently very difficult to navigate through their files.

Response: Thanks, yes we can see that with data from two papers, each consisting of two studies that the repository is pretty hard to understand. We have reorganised the folders to make it easier to navigate, and have added some instructions about which folders relate to which studies. 

P10 L 290

I commend the authors for mentioning this limitation, as it is both true and unfortunate that the majority of the literature relies on averaged scores. However, there have been previous attempts to address this limitation, so I recommend that the authors highlight these efforts. See, for example, Vincze et al., 2022, cited above.

Response: Thanks, we have added a reference to this paper and one other that looked at individual hits and misses. 

P 17 L 460

For a better understanding of the, a visual representation of the moving dots would be of great assistance.

Response: We have added a video to the OSF page as suggested. But the point of the manipulation was that the dots don’t look like they are moving, so it won’t look like they are moving in the video. 

P 19 L497

Can the authors additionally provide a number/code for their ethical approval and include it in the manuscript? I would also request that they submit the ethical approval when addressing the review procedure.

Response: Approval number added to the methods. 

P23 L 621. Can the authors discuss a potential neurological explanation for the inward-outward phenomenon?

Response: We think this is beyond the scope of the paper, and our area of expertise, but visual processing and attentional networks would be highly involved in both roles. 

P25 L 675

Can the authors address any limitations of the employed task or procedure? The limitation mentioned is quite general and applicable to a number of studies investigating Quiet Eye.

Response: We do mention some of the main limitations of the current work (in the Limitations section) in terms of the automated approach of QE calculation and the novice population. We think these are ones that are particularly relevant to the interpretation of the results, but are happy to acknowledge a particular limitation if the reviewer has something in mind that they think is important. However, all research approaches have inherent limitations (e.g., balancing experimental control against ecological validity) so we are not sure it is particularly useful to just list all the limitations we can think of. 

General note

I have a limited comprehension and application of Bayes factor and it's possible that my evaluation skills on this subject are poor. But, based on what I know, Bayes is fairly taxing when p is close to .05. I find the 'noisy eye' results (p =.04, BF = 318.07) to be somewhat inconsistent or whether there is an error in reporting. In addition, when considering the other results presented in the manuscript, ps and Bs "are essentially moving in the same direction." Could the authors explain how and why is this possible in this case?

Response: Yes, it is also our understanding that Bayes factors and p-values essentially converge for very small p-values but that they may differ a bit for values around p=.05. In the instance you mention the Bayesian evidence is obviously much stronger, but this is possible for smaller samples and results that are around this boundary. We checked and this is the correct result. This is the reason why we included both methods, because neither is ‘correct’ or infallible, but providing both helps to give the reader as much information as possible to draw a conclusion. 

General (minor) note

Perhaps this is just a personal preference, but I find that five citations for a single argument are quite a lot. In addition, certain sources are improperly cited in the manuscript (see for example P 8, l 217, l 220).

Response: We understand that multiple citations are not really needed, but we did it in some places to acknowledge research from a range of research groups. But we have removed some in a couple of places. As we have changed the referencing style to numbers the citation formatting issue should now be sorted out, thanks.

---

## [Decision Letter · Decision Letter 1]

14 Aug 2023

PONE-D-23-08839R1The functional role of visual information and fixation stillness in the quiet eyePLOS ONE

Dear Dr. Harris,

Thank you for submitting your manuscript to PLOS ONE. After careful consideration, we feel that it has merit but does not fully meet PLOS ONE’s publication criteria as it currently stands. Therefore, we invite you to submit a revised version of the manuscript that addresses the points raised during the review process. Please see my comments below under "Additional Editor Comments".

We look forward to receiving your revised manuscript.

Kind regards,

Nick Fogt

Academic Editor

PLOS ONE

Additional Editor Comments:

One of the reviewers (Reviewer #2) remains very concerned that there are fundamental issues related to data collection or analyses in the experiment, as evidenced by the large number of trials with no quiet eye periods. Can the authors reconcile their results with results of studies cited by the reviewer in which the quiet eye occurred with far more regularity? For example, is there something fundamentally different with the eye tracker, the methodology/threshold, the task, or the participant population that could account for these differences? Is it possible to vary the threshold or the method of detection of the quiet eye to see whether that makes a difference?

In addition to the critical issue raised by reviewer #2, please provide a reference or references around the "75ms change in quiet eye duration" used in the study.

Reviewers' comments:

Reviewer's Responses to Questions

**Comments to the Author**

1. If the authors have adequately addressed your comments raised in a previous round of review and you feel that this manuscript is now acceptable for publication, you may indicate that here to bypass the “Comments to the Author” section, enter your conflict of interest statement in the “Confidential to Editor” section, and submit your "Accept" recommendation.

Reviewer #1: (No Response)

Reviewer #2: (No Response)

2. Is the manuscript technically sound, and do the data support the conclusions?

Reviewer #1: Yes

Reviewer #2: No

3. Has the statistical analysis been performed appropriately and rigorously? 

Reviewer #1: Yes

Reviewer #2: Yes

4. Have the authors made all data underlying the findings in their manuscript fully available?

Reviewer #1: Yes

Reviewer #2: Yes

5. Is the manuscript presented in an intelligible fashion and written in standard English?

Reviewer #1: Yes

Reviewer #2: Yes

6. Review Comments to the Author

Reviewer #1: Thank you to the authors for addressing my comments. My last comment is regarding the selection of the 75ms change in quiet eye duration. Can the authors add some references for the reader to identify the previous literature mentioned in their edits?

Thank you for the opportunity to review this manuscript.

Reviewer #2: Review for PONE-D-23-08839R1

I appreciate your response to my earlier comments. However, your response regarding the lack of fixations in about half of the trials was insufficient. Henceforth, my focus will be solely directed towards this specific matter, with the intention of explaining my substantial concerns regarding the methodology adopted in the measurement of Quiet Eye.

Other studies have utilised more stringent spatial dispersion measures than those outlined in the present manuscript. The Tobii 2 glasses that were used in Vincze et al. (2022), for example, use an automated detection method (so fixations were not “manually coded”, as you suggest) and have a precision of 0.05° at 1.5 m (reaching 1.1 degree only in extremely poor lighting conditions). Despite these strict requirements for fixations, the measurement of QE was accomplished at a lower data loss rate in the literature (e.g., 1% in Vincze et al., 2022; 6,1 % Klosterman et al., 2018).

Moreover, in some of the aforementioned studies, the task's level of dynamism and speed were significantly higher (e.g., table tennis in Vincze et al.,) I mention all of this because we would expect higher data loss rates in a scenario with stricter precision and more/faster movement than those described in the present study.

Therefore, I do not understand how the high rate of data loss could be adequately explained by a strict definition of fixation alone. Consequently, I have concerns regarding any other factors that may have impacted the measurement process. As a result, I cannot recommend this work for publication in the present form as it does not sufficiently clarify this issue.

Klostermann, A., Kredel, R., & Hossner, E. J. (2018). Quiet eye and motor performance: The longer the better?. Journal of Sport and Exercise Psychology, 40(2), 82-91. doi: 10.1123/jsep.2017-0277

Tobii. (2022). Tobii Pro Glasses 2 Data Quality Test Report. Accuracy, precision, and data loss under controlled environment.

Vincze, A., Jurchis, R., & Iliescu, D. (2022). Quiet eye facilitates processing complex information in elite table tennis players. Visual Cognition, 30(7), 506-516.

7. PLOS authors have the option to publish the peer review history of their article (what does this mean?). If published, this will include your full peer review and any attached files.

Reviewer #1: No

Reviewer #2: No

---

## [Author Response · Author response to Decision Letter 1]

25 Sep 2023

Additional Editor Comments:

One of the reviewers (Reviewer #2) remains very concerned that there are fundamental issues related to data collection or analyses in the experiment, as evidenced by the large number of trials with no quiet eye periods. Can the authors reconcile their results with results of studies cited by the reviewer in which the quiet eye occurred with far more regularity? For example, is there something fundamentally different with the eye tracker, the methodology/threshold, the task, or the participant population that could account for these differences? Is it possible to vary the threshold or the method of detection of the quiet eye to see whether that makes a difference?

In addition to the critical issue raised by reviewer #2, please provide a reference or references around the "75ms change in quiet eye duration" used in the study.

Reviewer #1: Thank you to the authors for addressing my comments. My last comment is regarding the selection of the 75ms change in quiet eye duration. Can the authors add some references for the reader to identify the previous literature mentioned in their edits?

Thank you for the opportunity to review this manuscript.

Response: Added, thanks (line 160). 

Reviewer #2: Review for PONE-D-23-08839R1

I appreciate your response to my earlier comments. However, your response regarding the lack of fixations in about half of the trials was insufficient. Henceforth, my focus will be solely directed towards this specific matter, with the intention of explaining my substantial concerns regarding the methodology adopted in the measurement of Quiet Eye.

Other studies have utilised more stringent spatial dispersion measures than those outlined in the present manuscript. The Tobii 2 glasses that were used in Vincze et al. (2022), for example, use an automated detection method (so fixations were not “manually coded”, as you suggest) and have a precision of 0.05° at 1.5 m (reaching 1.1 degree only in extremely poor lighting conditions). Despite these strict requirements for fixations, the measurement of QE was accomplished at a lower data loss rate in the literature (e.g., 1% in Vincze et al., 2022; 6,1 % Klosterman et al., 2018).

Moreover, in some of the aforementioned studies, the task's level of dynamism and speed were significantly higher (e.g., table tennis in Vincze et al.,) I mention all of this because we would expect higher data loss rates in a scenario with stricter precision and more/faster movement than those described in the present study.

Therefore, I do not understand how the high rate of data loss could be adequately explained by a strict definition of fixation alone. Consequently, I have concerns regarding any other factors that may have impacted the measurement process. As a result, I cannot recommend this work for publication in the present form as it does not sufficiently clarify this issue.

Klostermann, A., Kredel, R., & Hossner, E. J. (2018). Quiet eye and motor performance: The longer the better?. Journal of Sport and Exercise Psychology, 40(2), 82-91. doi: 10.1123/jsep.2017-0277

Tobii. (2022). Tobii Pro Glasses 2 Data Quality Test Report. Accuracy, precision, and data loss under controlled environment.

Vincze, A., Jurchis, R., & Iliescu, D. (2022). Quiet eye facilitates processing complex information in elite table tennis players. Visual Cognition, 30(7), 506-516.

Response: Thanks, yes we appreciate that there are a lot of zero values and understand the reviewer digging deeper into this issue. We are confident that the zero values are a legitimate aspect of the data and not some kind of error. To expand on our previous response to this issue:

1. We used a novice population who have typically short quiet eye durations. Novice participants can be quite bad at controlling their attention and may often have no quiet eye at all. The comparison to the Vincze et al. paper with elite sports people who have much better attentional control is therefore somewhat problematic. Indeed, the most consistent finding in QE-related research is this between group difference in QE duration between experts and novices (See Lebeau et al., 2016 meta analysis). As the reason for this difference is related to better attentional control, it is not unexpected that QE detection would be lower in a non-experienced sample population who do not know where to best focus attention (either in terms of duration or location). 

2. You mention the spatial dispersion values in other studies, but it is not simply the spatial dispersion criteria that is the issue. It is the fact that because we are using VR we are able to precisely locate both the location of the ball in 3D space and the intersection point of the gaze vector in 3D space, so in addition to requiring the gaze to be stable within a certain spatial dispersion, this must also occur with the gaze staying on the target location (the ball) as well. Our point about the manual coding is that there is a lot more ‘flex’ in that method which allows the fixation to continue if gaze moves slightly, but still remains (functionally) near the ball. 

Whereas, with our automated method the determination of this position is more stringent and the quiet eye period would end when gaze and object do not overlap. The automated method in VR deals with issues around the vestibulo-ocular reflex that head mounted eye trackers cannot deal with without some additional head tracking mechanism (i.e., eyes might be stable in relation to the head but point of gaze might be moving, which would be a continuation of a fixation for head mounted eye trackers, but not in VR). The combination of the strict fixation detection algorithm, with the strict matching of the gaze vector to the ball position is the reason why our method resulted in more zero durations than most. We do acknowledge in the paper that our values are probably not comparable to other studies for this reason. 

3. A key point that we would like to reiterate as researchers who have adopted both methods, is that zero quiet eye durations are not an error or an indication that something has gone wrong. As such the term ‘data loss’ is problematic. We contend that it is not that we were unable to detect gaze behaviour that was happening, but that this behaviour was not present to the degree of precision that we have set. 

Additionally, it is incorrect to say: ‘regarding the lack of fixations in about half of the trials’, because, what we show is that there just isn’t a stable fixation in the right place at the right time to register as our definition of a quiet eye. A zero quiet eye is a totally legitimate behaviour and may mean the participant has focused near the ball, but not exactly on it, or they looked at it but their eye was not stable. Or that they barely looked at it. All of these things result in a zero quiet eye being recorded but are not ‘data loss’. Also, this outcome occurred on 35% of trials, which means that total novices do not execute a quiet eye on 1 in 3 trials, which seems entirely reasonable if the quiet eye is a measure of expertise and optimal attentional control. 

4. While we appreciate the comments and pointers to the Vincze et al. paper, this is not the most appropriate comparator for a couple of reasons. Firstly, it is not a self-paced task, so the quiet eye is a tracking gaze of a moving target. This is in itself a problem when it comes to defining QE across tasks (and something our team has also wrestled with in the past when computing QE variables for tracking tasks). Secondly, in the Vincze et al. paper it is not stated that the fixation was actually matched to the ball’s location (From the Methods section of the paper – “QE was operationalized as the final fixation before the athlete initiates a critical movement. The fixation has to last at least 100 ms and not deviate more than 3°”). Any steady gaze (of >80ms) to any location would have been calculated as a QE, which is a much less stringent than our definition which reflects the fixation to a specific location in time and space. It is almost inevitable that there would have been some kind of fixation before movement initiation in the Vincze study, so there would be very few zero values (despite this being a more dynamic task). However, from the description provided in the Methods of the paper, this fixation could have been on the coach’s face, body, arm, the ball, the table, or somewhere in the space between. If instead the authors had matched the gaze vector to the actual location of the ball in 3D space (very difficult with head mounted eye trackers but easy in a VR headset) this would have resulted in a much stricter criterion for the quiet eye and likely many more zero values. 

5. The quiet eye calculation method that we used is indeed much more similar to the Klosterman paper, but this comparison is still somewhat problematic. Firstly, Klosterman et al experimentally cued the quiet eye durations of the participants using auditory tones and varying stimulus onset delays. So, they were controlling the durations, which is not comparable to what we did. Secondly, they trained the participants to do this, so again they were not really novices in the throwing task. In addition, throwing a ball is a common skill, whereas golf is much more specialised and unnatural and we specifically recruited novices. Thirdly, again this is not a comparable task because the target appears at a variety of locations, so you can’t really throw a projectile at it without looking at it. By contrast, during putting you know the location of the target, so a quiet eye becomes an attentional “choice” and might not be activated (or even understood as a solution to performance feedback). Indeed, as we showed in this very study, you can putt effectively without having to look at the ball at all (occlusion conditions). So, it is entirely possible that participants would not have had a quiet eye sometimes (especially novices). 

Given these points, we hope the reviewer will consider that these two studies are really not comparable to what we have done, and that the differences we have pointed out are where the much higher percentage of zero values have come from. Most quiet eye studies present averaged quiet eye values and have not provided the raw data in open repositories, so it is hard to know how often novice performers have zero quiet eye values in these studies. To reiterate, zero values are not ‘data loss’, they are just when a participant did not execute a stable fixation on the ball at the right time, something that would be a common occurrence in our novice population.

---

## [Editor Report · Decision Letter 2]

27 Sep 2023

PONE-D-23-08839R2The functional role of visual information and fixation stillness in the quiet eyePLOS ONE

Dear Dr. Harris,

Thank you for submitting your manuscript to PLOS ONE. After careful consideration, we feel that it has merit but does not fully meet PLOS ONE’s publication criteria as it currently stands. Therefore, we invite you to submit a revised version of the manuscript that addresses the points raised during the review process.

We look forward to receiving your revised manuscript.

Kind regards,

Nick Fogt

Academic Editor

PLOS ONE

Journal Requirements:

Additional Editor Comments:

Thank you for the extensive response to the reviewer's comments. Can you please indicate what has been changed/added in the manuscript to address the reviewer concerns (perhaps by highlighting these changes or by using a different font). Much of this discussion can perhaps be incorporated in a separate section in the paper perhaps entitled "methodological considerations with the quiet eye" or something like that. The journal readership will benefit from this discussion.

---

## [Author Response · Author response to Decision Letter 2]

9 Oct 2023

Additional Editor Comments:

Thank you for the extensive response to the reviewer's comments. Can you please indicate what has been changed/added in the manuscript to address the reviewer concerns (perhaps by highlighting these changes or by using a different font). Much of this discussion can perhaps be incorporated in a separate section in the paper perhaps entitled "methodological considerations with the quiet eye" or something like that. The journal readership will benefit from this discussion.

Response: Thanks. As you saw from the previous set of responses we think the reviewers comments were based on a limited understanding of this literature in this area, so did not warrant any substantive changes to the manuscript. We tried to justify this in our detailed responses in which we showed that the reviewer’s comparisons to other studies were not warranted. 

In response to this additional request, we have added a paragraph to the discussion to outline some of these methodological issues and how our paper compares to previous work in this area (lines 658-667).

---

## [Editor Report · Decision Letter 3]

24 Oct 2023

The functional role of visual information and fixation stillness in the quiet eye

PONE-D-23-08839R3

Dear Dr. Harris,

We’re pleased to inform you that your manuscript has been judged scientifically suitable for publication and will be formally accepted for publication once it meets all outstanding technical requirements.

Kind regards,

Nick Fogt

Academic Editor

PLOS ONE
---

## [Editor Report · Acceptance letter]

27 Oct 2023

PONE-D-23-08839R3 

The functional role of visual information and fixation stillness in the quiet eye 

Dear Dr. Harris:

I'm pleased to inform you that your manuscript has been deemed suitable for publication in PLOS ONE. Congratulations! Your manuscript is now with our production department. 

Kind regards, 

on behalf of

Dr. Nick Fogt 

Academic Editor

PLOS ONE